# Zika virus RNA structure controls its unique neurotropism by bipartite binding to Musashi-1

Xiang Chen[1,6], Yan Wang [2,6], Zhonghe Xu [2,6], Meng-Li Cheng[1], Qing-Qing Ma[1], Rui-Ting Li[1], Zheng-Jian Wang[1], Hui Zhao[1], Xiaobing Zuo[3], Xiao-Feng Li[1], Xianyang Fang [2,4] ✉ & Cheng-Feng Qin [1,5] ✉

Human RNA binding protein Musashi-1 (MSI1) plays a critical role in neural progenitor cells (NPCs) by binding to various host RNA transcripts. The canonical MSI1 binding site (MBS), A/GU$_{(1-3)}$AG single-strand motif, is present in many RNA virus genomes, but only Zika virus (ZIKV) genome has been demonstrated to bind MSI1. Herein, we identified the AUAG motif and the AGAA tetraloop in the Xrn1-resistant RNA 2 (xrRNA2) as the canonical and non-canonical MBS, respectively, and both are crucial for ZIKV neurotropism. More importantly, the unique AGNN-type tetraloop is evolutionarily conserved, and distinguishes ZIKV from other known viruses with putative MBSs. Integrated structural analysis showed that MSI1 binds to the AUAG motif and AGAA tetraloop of ZIKV in a bipartite fashion. Thus, our results not only identified an unusual viral RNA structure responsible for MSI recognition, but also revealed a role for the highly structured xrRNA in controlling viral neurotropism.

Zika virus (ZIKV) infection of pregnant women can cause fetal microcephaly[1]. Interrogation of ZIKV-infected human brain organoids and mouse models demonstrated that ZIKV infection hampers the development of neural progenitor cells (NPCs)[2–5], which may account for microcephaly in human fetuses or newborn babies. More importantly, ZIKV mainly targets NPCs in the developing brain and replicates with high efficiency in NPCs, whereas mature neurons are less sensitive to ZIKV infection[2,6,7]. This characteristic of ZIKV stands in stark contrast with other known flavivirus members[6].

RNA-binding protein Musashi homolog 1 (MSI1) is an evolutionarily conserved RNA-binding protein originally discovered in the central nervous system (CNS)[8]. It is highly enriched in undifferentiated neural stem cells (NSCs) and neural progenitor NPCs, but is massively downregulated following the successive progression of neurogenesis[8–10]. MSI plays a critical role in maintaining the self-renewal of NSCs by binding to the 3′ untranslated regions (UTRs) of specific target mRNAs[11–14]. Knock out of MSI1 in mice resulted in abnormal brain development and reduced multi-potency of CNS stem cells[15]. MSI1 comprises two RNA recognition motifs (RRM1 and RRM2), connected by a short linker in the N-terminal region followed by an intrinsically disordered C-terminal region. The MSI1 binding site (MBS) has been well characterized as single-stranded RNA motifs containing the consensus sequence A/GU$_{(1-3)}$AG[11,16–19], and canonical MBSs in various host RNA transcripts encoding proteins with diverse biological functions have been validated[11,20–22].

The ZIKV genome contains a single open reading frame flanked by 5′ and 3′ UTRs. The highly structured 3′ UTR of 429 nucleotides in length is composed of two xrRNAs (Xrn1-resistant RNA: xrRNA1 and xrRNA2), two dumbbells (DBs, DB1 and DB2), and a 3′ stem-loop (SL) subdomain[23], and is important for viral replication and

[1]Department of Virology, State Key Laboratory of Pathogen and Biosecurity, Beijing Institute of Microbiology and Epidemiology, AMMS, Beijing 100071, China. [2]Beijing Advanced Innovation Center for Structural Biology and Beijing Frontier Research Center for Biological Structure, School of Life Sciences, Tsinghua University, Beijing 100084, China. [3]X-ray Science Division, Argonne National Laboratory, Lemont, IL 60439, USA. [4]Key Laboratory of RNA Biology, Institute of Biophysics, Chinese Academy of Sciences, Beijing 100101, China. [5]Research Unit of Discovery and Tracing of Natural Focus Diseases, Chinese Academy of Medical Sciences, Beijing 100071, China. [6]These authors contributed equally: Xiang Chen, Yan Wang, Zhonghe Xu. ✉e-mail: fangxy@ibp.ac.cn; qincf@bmi.ac.cn

pathogenesis[24,25]. Recently, Chavali et al. [26] demonstrated that MSI1 directly binds to the 3´ UTR of the ZIKV genome and promotes viral replication in cells expressing MSI1. Sequence analysis and computational prediction[27–29] have predicated multiple putative MBSs (pMBSs) in the 3´ UTR of ZIKV, although none of these pMBSs have been validated. More importantly, these pMBSs are distributed in a large number of viruses across different families with different properties, especially mosquito-borne flaviviruses (MBFV)[26,29]. However, ZIKV is the only known pathogen linked to human neurodevelopmental disorders. This unusual phenotype questions the canonical rule for MSI1 binding, and the authentic viral RNA targets of MSI1 demands urgent investigation.

In this study, using combined technology platforms, we identified two MBSs in the xrRNA2 of the ZIKV 3´ UTR, and clarified their crucial roles during ZIKV replication in NPCs. Structural analysis using integrated methods demonstrated a unique bipartite binding mode for both xrRNA2 and MSI1. Our study not only characterized the evolutionally conserved RNA elements in the ZIKV genome that cooperatively regulate viral replication in diverse cells, but also provides a novel example showing how a virus adopts its RNA primary sequence and tertiary structure to hijack a host protein to facilitate its specific tropism and life cycle.

## Results

### The AUAG motif in ZIKV xrRNA2 is the authentic RNA target of MSI1

Human MSI1, constituted by RRM1 and RRM2, directly binds to the ZIKV 3´ UTR[26]. To characterize the role of each RRM of MSI1 in RNA-protein interactions, isothermal titration calorimetry (ITC) assays were performed using the tandem RRM12 (20–191) and the individual RRM1 (20-95) and RRM2 (108–191) sequences of MSI1 and the ZIKV 3´ UTR (Fig. S1a, b). As expected, the tandem RRM12 shows obvious binding to the ZIKV 3´ UTR with a binding affinity of 2.45 μM, while the individual RRM1 or RRM2 subdomains failed to bind with the ZIKV 3´ UTR (Fig. S1c). This result suggests that both the RRM1 and RRM2 subdomains of MSI1 are required for cooperative binding to the ZIKV 3´ UTR.

Based on the canonical A/GU$_{(1-3)}$AG rule, three pMBSs (pMBS1-3) have been in silico mapped to the xrRNA2, DB12, and 3´ SL subdomains of the ZIKV 3´ UTR, respectively (Fig. 1a). In order to clarify the role of each pMBS, the individual subdomains of xrRNA2, DB12 and the 3´ SL harboring pMBS1, pMBS2, and pMBS3, respectively, were constructed, and their interactions with MSI1 RRM12 were measured by ITC. Remarkably, only xrRNA2 showed high binding affinity (1.76 μM) to MSI1; whilst the binding affinity of the 3´ SL/ DB12 and MSI1 was weak (Fig. 1b). Furthermore, three mutants of the ZIKV 3´ UTR, named as pMBS1m, pMBS2m, and pMBS3m, respectively, were constructed, in which the respective pMBSs were disrupted without altering the 3´ UTR secondary structure (Fig. 1c). Small angle X-ray scattering (SAXS) data revealed that the mutations didn't alter the overall structure of the ZIKV 3' UTR (Fig. S1d–f and Table S1). ITC assays showed that only the mutations contained in pMBS1m significantly weakened MSI1 binding, whilst the mutated pMBS2m and pMBS3m structures had similar binding affinity as the wild type (WT) 3' UTR (Fig. 1d and Table S2). RNA pull-down assays further confirmed that the mutations in pMBS1m significantly decreased the binding to MSI1 which was not the case for either pMBS2m or pMBS3m (Fig. 1e). As a control, we also detected whether these three mutations affect the interaction of 3' UTR with Fragile X Mental Retardation Protein (FMRP), which was previously reported to interact with ZIKV 3´ UTR[30]. As expected, there was no observed difference in FMRP binding between WT 3'UTR RNA and the three mutant RNAs (Fig. 1e), suggesting the pMBS1m mutation harbors a specific effect on MSI1 binding. Together, these results demonstrate that pMBS1, the AUAG motif, in xrRNA2 is crucial for MSI1 binding.

To further determine the function of pMBS1 during ZIKV infection, the corresponding pMBS1m mutation was engineered into the infectious clone of ZIKV strain FSS13025[31] and the corresponding mutant virus (pMBS1m) rescued (Fig. S2a). As expected, ZIKV WT and pMBS1m exhibited similar plaque morphologies (Fig. S2b) and replication kinetics in BHK-21 cells, suggesting that the pMBS1m mutation does not affect viral replication in the absence of MSI1 (Fig. 1f). In addition, the northern blot result showed that the pMBS1m mutation has no influence on the production of sfRNAs (Fig. S2c). More importantly, in BHK-21 cells that transiently expressed MSI1, RNA immunoprecipitation (RNA-IP) assays showed a remarkable reduction in the binding of MSI1 to viral RNA following ZIKV pMBS1m infection, compared to the WT virus (Fig. 1g). In contrast, the control protein FMRP showed similar binding ability to WT ZIKV and pMBS1m ZIKV RNA (Fig. S2d). Consistently, the co-localization of MSI1 and viral RNA was significantly reduced in ZIKV pMBS1m infected cells (Fig. 1h). Taken together, our results indicated that the AUAG motif in the ZIKV xrRNA2 is the authentic RNA target of MSI1.

### The pMBS1 promotes ZIKV replication in an MSI1-dependent manner

We further sought to determine the function of pMBS1 during ZIKV replication. A panel of human cell lines that endogenously express MSI1 (Fig. S2e) was subjected to infection with ZIKV and the corresponding pMBS1 mutant virus. Compared with the WT ZIKV, the replication of pMBS1m was significantly decreased over the course of infection in hNPCs (Fig. 2f, g). Similarly, the pMBS1m mutant virus replicated to lower levels compared to the WT virus in U251 and SH-SY5Y cells (Fig. S2c, d). These results showed pMBS1 promoted ZIKV replication in a cell-specific manner.

A BHK-21-MSI1 cell line with stable MSI1 expression and the corresponding control cell line (BHK-21-Ctrl) were generated by lentivirus transduction (Fig. S2e) and the replication kinetics of the WT and pMBS1m mutant virus were compared. Similarly to previous results (Fig. 1f), the pMBS1m and WT viruses showed similar replication efficiency in BHK-21-Ctrl cells lacking endogenous MSI1. In contrast, the growth of the pMBS1m virus was significantly decreased, compared with the WT virus in BHK-21-MSI1 cells that stably express MSI1 (Fig. 2c and Fig. S2h). Furthermore, knockdown of MSI1 in hNPCs significantly down-regulated ZIKV E protein expression (Fig. 2d) as well as viral RNA amounts (Fig. 2e) after WT ZIKV infection, whereas there was no impact on the growth of pMBS1m.

The pMBS1 mutation was also introduced into a ZIKV replicon (ZIKVrep) containing a *Renilla* luciferase reporter gene (Fig. S2i), resulting in the mutant ZIKV replicon Rep-pMBS1m. Rep-pMBS1m replicated as efficiently as WT in BHK-21-Ctrl cells but with less efficiency in BHK-21-MSI1 cells (Fig. S2j, k). Furthermore, Rep-pMBS1m displayed sharply decreased RNA replication in hNPCs which normally express MSI1 (Fig. S2l). Collectively, the results demonstrate that pMBS1 positively regulates ZIKV replication in an MSI1-dependent and cell-specific manner.

### The AGAA tetraloop fold in xrRNA2 is critical for MSI1 binding

The structural basis of MSI1 binding to the identified MBS of ZIKV was then investigated. The structure of ZIKV xrRNA1 has been determined with a ring-like 3D structure[32]. The homology-derived secondary structure of xrRNA2 was predicated to include a three-way junction formed by P1, P2, and P3 and an additional P4 helix (Fig. 3a, left). As the corresponding P2 and P4 structures in ZIKV xrRNA1 were altered to stabilize the construct for crystallization[32], we also constructed a similar xrRNA2 mutant (xrRNA2 P2M/P4M), in which the apical loops capping the P2 and P4 stems were simultaneously mutated to produce a GAAA tetraloop (Fig. 3a, right). Unexpectedly, the P2M/P4M mutant completely lost the ability to bind MSI1 (Fig. 3b). To investigate the lack of MSI1 binding further, two xrRNA2 mutants were constructed in

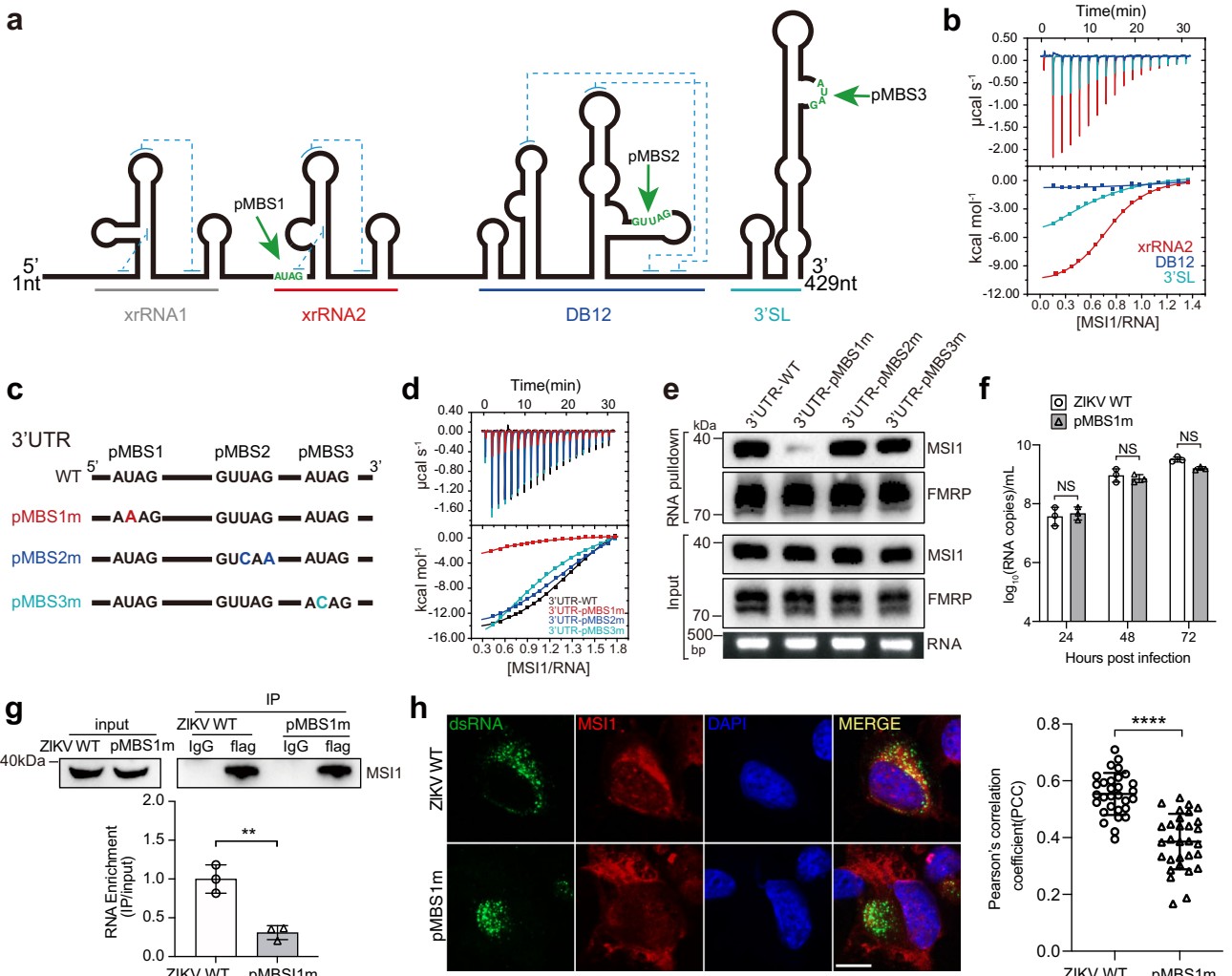

**Fig. 1 | The pMBS1 in ZIKV xrRNA2 is responsible for MSI1 binding. a** Putative MSI1 binding sites (pMBS) in the 3′ UTR of ZIKV. The pseudoknots are indicated with blue dashed lines. **b** The ITC profiles of RRM12 to the individual xrRNA2, DB12, and 3′ SL sequences. **c** Schematic showing mutations in the putative MSI1 binding sites in the 3′ UTR of ZIKV. **d** The ITC profiles of RRM12 to the ZIKV 3′ UTR WT, pMBS1m, pMBS2m and pMBS3m. **e** RNA pull-down assays performed with the WT or single pMBS mutant (1-3) 3′ UTRs of ZIKV. In vitro transcribed biotinylated RNAs were incubated with cell extracts of BHK-21 cells transfected with flag-MSI1, and RNA-protein complexes were captured on streptavidin beads. Representative Western blots probed with antibody against MSI1 are shown together with corresponding protein and RNA inputs. Similar result was repeated independently in 3 times. **f** ZIKV WT and pMBS1m viral RNA copies in culture supernatants of BHK-21 cells (MOI = 0.1). Data are the mean ± SD. $n$ = 3 independent experiments. Two-way ANOVA, NS, not significant. **g** RNA-IP analysis from WT or pMBS1m infected BHK-21

cells transfected with flag-MSI1. Western blot shows immunoprecipitations (IPs) by immunoglobulin G (IgG) and flag antibodies. Input (5%) represents whole-cell extract. Western blot was probed with antibodies against flag. The bound viral RNA from the IP was analyzed by qRT-PCR. Graph below shows qRT-PCR performed on bound RNA from IP. RNA-IP values are presented as the ratio to the input after subtraction of the IgG background. Data are mean ± SD. $n$ = 3 independent experiments. Two-sided Student's $t$ test, **$P$ < 0.01. ($P$ = 0.0042). **h** Confocal microscopy of BHK-21 cells transduced with MSI1-expressing lentiviruses (MOI = 10) and infected with ZIKV WT or pMBS1m (MOI = 1) for 24 h. Fixed cells were immunostained with anti-dsRNA and anti-MSI1 antibodies. Cell nuclei were counterstained with DAPI. Scale bar, 10 μM. Graph on the right displays the Pearson's correlation coefficient (PCC), mean ± SD was calculated from 30 cells in each group. Two-sided Student's $t$ test. ****$P$ < 0.0001. ($P$ < 0.0001). Source data are provided as a Source Data file.

which the apical loops capping the P2 or P4 stems were separately replaced with GAAA tetraloops (P2M and P4M) and subjected to MSI1 binding analysis by ITC (Fig. S3a and Table S2). The P2M mutant lost the capacity to bind MSI1 which was preserved for the P4M mutant (Fig. 3c), suggesting that the P2 tetraloop, but not P4, is involved in MSI1 binding. This was further demonstrated by the deletion of the whole P4 (Δ4 P) structure, which did not disturb MSI1-xrRNA2 binding (Fig. S3a, Fig. 3c and Table S2). Additionally, SAXS analysis showed that the overall structure of the P2M mutant is similar to that of the WT xrRNA2 (Fig. S3b−d and Table S1). These results suggested that the apical P2 tetraloop in xrRNA2 functions as a non-canonical MBS.

The AGAA loop ending the P2 stem of xrRNA2 belongs to the AGNN-type tetraloop, in which a non-canonical A-A base pair is formed

closing the helix, and the intermediate guanine and adenine are flipped out[33,34]. The GAAA tetraloop is a typical GNRA-type tetraloop, which also has a non-canonical G-A base pair closing the helix, but the three adenines are stacked with the 3′ guanine[34] (Fig. 3d). To clarify whether the tetraloop fold is critical for MSI binding, several xrRNA2 mutants with modified P2 apical loops were constructed and analyzed for their ability to bind MSI1 (Fig. S3e). The AGNN-type mutants, in which the AGAA loop was replaced with AGCU (P2-AGCU) or AGUU (P2-AGUU), retained strong MSI1 binding affinities. By contrast, the GNRA-type mutants of xrRNA2, in which the AGAA loop was replaced with GAGA (P2-GAGA) or GCAA (P2-GCAA), showed significantly decreased MSI1 binding affinities (Fig. 3e and Table S2). These results demonstrated that the P2 tetraloop fold of xrRNA2 is critical for MSI1 binding.

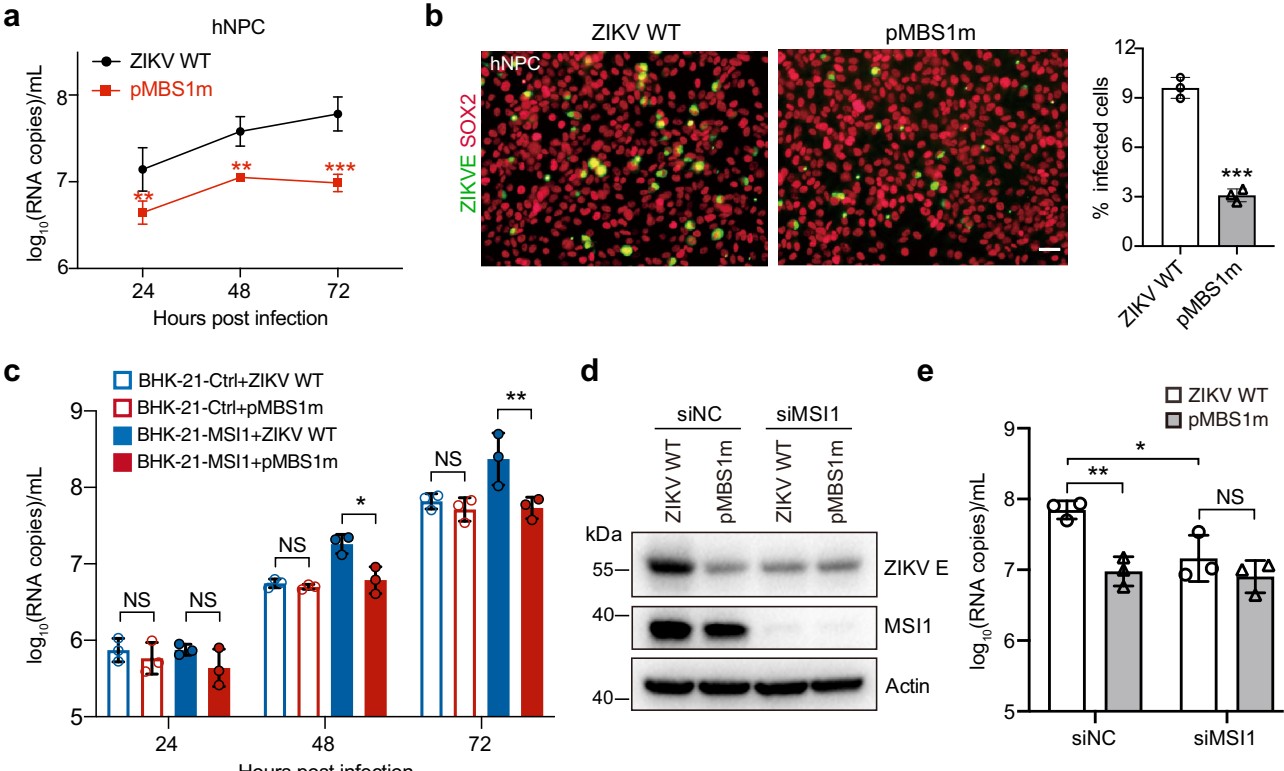

**Fig. 2 | The pMBS1m mutation attenuates ZIKV replication in cells with MSI1 expression. a** hNPCs were infected with the ZIKV WT or pMBS1m viruses (MOI = 1), and the culture supernatants were harvested at the indicated time points for detection of viral RNA copies by qRT-PCR. Data are mean ± SD. $n$ = 3 independent experiments. Two-way ANOVA, **$P < 0.01$, ***$P < 0.001$. (24 h $P$ = 0.0083; 48 h $P$ = 0.0054; 72 h $P$ = 0.0002). **b** The expression of viral envelope protein at 48 h after infection from (a) was detected by immunostaining. Scale bar, 50 um. Data are mean ± SD. Two-sided Student's $t$ test, ***$P < 0.001$. ($P$ = 0.0001). **c** BHK-21-Ctrl and BHK-21-MSI1 cells were infected with the ZIKV WT or pMBS1m viruses (MOI = 0.01), and culture supernatant was harvested at the indicated time points for detection of viral RNA copies by qRT-PCR. Data are mean ± SD. $n$ = 3 independent experiments.

Two-way ANOVA, *$P < 0.05$, **$P < 0.01$, NS not significant. (48 h: BHK-21-MSI1-WT vs BHK-21-MSI1 pMBS1m, $P$ = 0.0438; 72 h: BHK-21-MSI1-WT vs BHK-21-MSI1 pMBS1m, $P$ = 0.0021). **d**, **e** hNPCs were treated with negative control siRNA (siNC) and MSI1 siRNA (siMSI1) for 36 h, and then infected with the ZIKV WT or pMBS1m viruses (MOI = 3), and at 48 h after infection, viral envelope and MSI1 in cell lysates was detected by Western blot (d), culture supernatant were harvested for detection of viral RNA copies by qRT-PCR (e). Data are mean ± SD. $n$ = 3 independent experiments. Two-sided Student's $t$ test, *$P < 0.05$, **$P < 0.01$, NS not significant. (siNC WT vs siNC pMBS1m, $P$ = 0.0035; siNC WT vs siMSI1 WT, $P$ = 0.0275). Source data are provided as a Source Data file.

The effects of structure-based mutation of P2M on MSI1 binding in vitro led us to predict that the same mutation would alter MSI1 binding ability during infection. Therefore, we produced a P2M mutant ZIKV, in which the xrRNA2 P2 was replaced with the sequence 5′ CC-GAAA-GG 3′ and investigated the effects of the mutation on virus replication. The plaque and sfRNA production phenotypes of the WT and P2M viruses were similar (Fig. S3f, g), and both viruses replicated equally well in BHK-21 cells that lacked endogenous MSI1 expression (Fig. S2e). RNA-IP was performed using lysates prepared from ZIKV WT or P2M infected BHK-21 cells transfected with MSI1, which revealed a significantly reduced interaction between MSI1 and the P2M RNA compared to the WT virus RNA (Fig. 4a); While the interaction of the control protein FMRP with P2M RNA is similar to that of WT virus RNA (Fig. S3i). Consistently, there was less colocalization observed between MSI1 and double-stranded RNA in ZIKV P2M-infected cells compared to WT virus-infected cells (Fig. 4b). These data confirmed the important role of the xrRNA2 P2 tetraloop fold in mediating in vitro and in vivo MSI1 binding.

More importantly, the P2M mutation significantly reduced virus propagation and replicon RNA synthesis in BHK-21-MSI1 cells compared to the WT equivalents, whereas no differences were observed in BHK-21-Ctrl cells (Fig. S3g–j). Furthermore, in hNPCs, ZIKV P2M showed markedly lower levels of viral RNA and E protein expression compared with WT virus (Fig. 4c, d), and the Rep-P2M replicon

produced much less luciferase signal than the WT replicon at 48 h post-transfection (Fig. S3k). Collectively, these data demonstrated that the AGAA tetraloop of xrRNA2, like pMBS1, regulates ZIKV replication in an MSI1-dependent manner, representing a non-canonical MBS.

To rule out the possibility that attenuation of P2M ZIKV was owing to a non-specific effect of P2 sequence change, we produced an additional mutant ZIKV, P2-AGCU (Fig. S4a), in which the xrRNA2 P2 was replaced with the sequence 5′ CG-AGCU-CG 3′, keeping the AGNN-type of P2 unchanged. P2-AGCU ZIKV and WT ZIKV replicated equally well in BHK-21 cells and hNPCs (Fig. S4b–d), which was consistent with that the P2-AGCU mutation did not affect MSI1 binding affinity (Fig. 3e). These data reinforced the conclusion that the contribution of xrRNA2 P2 to ZIKV replication depends on its AGNN-type structure.

**The AGNN-type tetraloop of xrRNA2 is only present in ZIKV**
Considering the integral role of the identified MBSs in the ZIKV genome, we determined whether similar MBSs were also present in the 3′ UTRs of other viruses by sequence alignments and RNA structure predictions. Intriguingly, a large number of viruses, including coxsackievirus B3 (CVB3), hepatitis C virus (HCV), human rhinovirus 14 (HRV14), Norwalk virus (NV), western equine encephalitis virus (WEEV), Sindbis virus (SINV), severe acute respiratory syndrome coronavirus 2 (SARS-CoV-2) and murine hepatitis virus (MHV), contain pMBS sequences, with SARS-CoV-2 and MHV containing

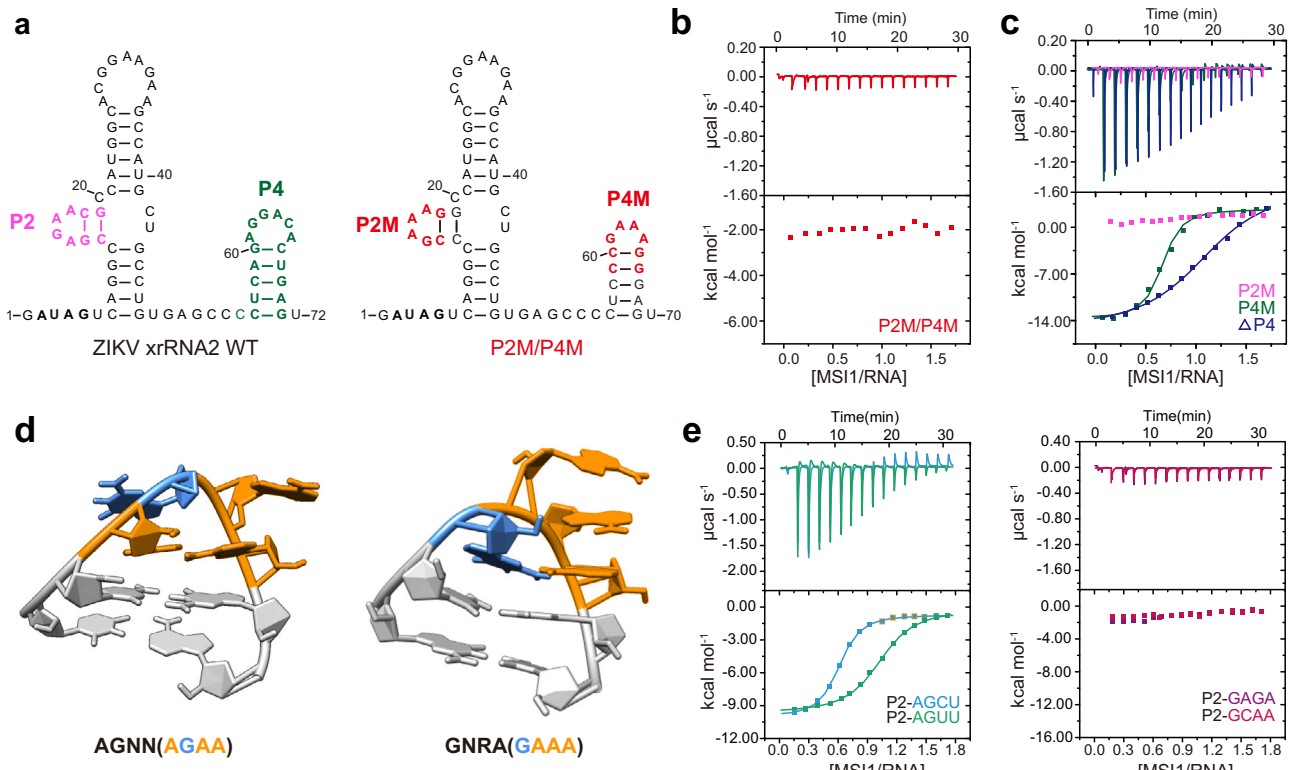

**Fig. 3 | xrRNA2 P2 tetraloop fold is critical for MSI1 binding. a** Secondary structure of ZIKV xrRNA2 and the xrRNA2 mutant P2M/P4M. **b** ITC profiles of RRM12 to xrRNA2 P2/P4M. **c** The ITC profiles of RRM12 to xrRNA2 mutants P2M, P4M and △P4. **d** NMR structures of tetraloops of AGNN and GNRA. **e** ITC profiles of RRM12 to xrRNA2 P2 loop mutants of ZIKV xrRNA2. Mutants with AGNN-type tetraloop include P2-AGCU and P2-AGUU. Mutants with a GNRA-type tetraloop include P2-GAGA and P2-GCAA.

seven and five pMBSs in their 3′ UTRs respectively (Fig. S5a). As for flaviviruses, the vast majority of them contain the pMBS sequence (Fig. S5b), while none of these viruses harbor the A/GUAG and AGNN-type tetraloop simultaneously, except for ZIKV (Fig. 5b). Some flaviviruses contain the same AUAG motif in their xrRNAs, but their P2 loops don't belong to the AGNN-type. Further analysis of MSI1 binding showed that the 3′ UTRs from non-ZIKV flaviviruses including MVEV, USUV, JEV, YFV, and DENV exhibited no binding affinity to MSI1 (Fig. 5c). We then tested the impact of MSI1 on the replication of these (mosquito-borne flaviviruses) MBFVs. As expected, MSI1 knockdown by RNAi in hNPCs reduced ZIKV replication, while the replication of YFV, JEV, WNV, DENV2, and DENV4, were not affected by MSI1 knockdown (Fig. 5d and e). More importantly, phylogenetic analysis revealed that ZIKV 3′ UTRs clustered in distinct clades: American, Asian, and African, but the AUAG motif, as well as the AGAA tetraloop, are both evolutionarily conserved in all ZIKV isolates (Fig. S6). Collectively, these results indicated that the unique AGAA tetraloop of xrRNA2 distinguishes ZIKV from other flaviviruses.

### The MSI1 binding element of ZIKV xrRNA2 confers MSI1 binding ability to other flaviviruses

We next investigated whether the MSI1 binding element alone is sufficient to endow other flaviviruses neurotropism like ZIKV. As shown above, DENV4 xrRNA does not contain A/GUAG sequence nor AGNN-type tetraloop (Fig. 5b), and has no MSI1 binding ability (Fig. 5c). We then transplanted the pMBS sequence and AGAA tetraloop of ZIKV xrRNA2 to the corresponding positions of DENV4 xrRNA. As shown in Fig. 6A, the $A_3$ and $A_{13}C_{14}U_{15}U_{16}G_{17}U_{18}$ of DENV4 xrRNA were mutated into $U_3$ and $G_{13}A_{14}G_{15}A_{16}A_{17}C_{18}$, respectively, generating a DENV4 xrRNA mutant (DxM), to endow it with similar configuration to ZIKV xrRNA2. Then we detected the binding between DxM RNA and MSI1 by

ITC assay. MSI1 shows high affinity to DxM xrRNA with Kd at 2.5 μM, which is similar to that of ZIKV xrRNA2 (Fig. 6b, left). The binding between DENV4 3′UTR mutant and MSI1 is also enhanced (Fig. 6b, right). We subsequently introduced this mutant to a DENV4 814669 strain infectious clone[24], and constructed an xrRNA mutated DENV4 virus (DENV4 DxM). BHK-21 cells or hNPCs were infected with WT DENV4 or DxM and analyzed viral replication. DxM infection measured by viral RNA replication was similar to WT in BHK-21 cells (Fig. 6c). More importantly, DxM exhibited enhanced replication ability in hNPCs (Fig. 6d, e). Thus, transplantation of MSI1 binding element of ZIKV xrRNA2 could confer other flaviviruses enhanced replication in MSI1 expressing cells.

### MSI1 RRM12 utilize two different RNA binding interfaces to bind to xrRNA2

The two RRMs of MSI1 adopts a canonical β1–α1–β2–b3–α2–β4 topology, and the conserved aromatic side chains (termed the RNP consensus sequence) that are essential for RNA binding are located on the β1 (RNP2) and β3 (RNP1) strands. However, the AGAA tetraloop doesn't belong to the conserved MSI1 binding sequence, and how the two homogeneous RRMs bind to two different RNA sequences is still unknown. To explore whether MSI1 utilizes the AGAA tetraloop with the canonical RNPs, we determined the interaction interfaces within RRM12 using Hydrogen-Deuterium Exchange mass spectrometry (HDX-MS). Experiments were performed for RRM12 alone (*apo*-form) or in the presence of a 2-fold molar excess of xrRNA2 (*holo*-form) (Table S5). A total of 98 and 102 peptides from the *apo*-form and *holo*-form RRM12, respectively, were identified and characterized across all HDX time points, representing a high sequence coverage (95%) (Fig. S7a, Supplementary Data 1). In the *apo* state, most peptides exhibited high solvent accessibility with relative deuterium uptake

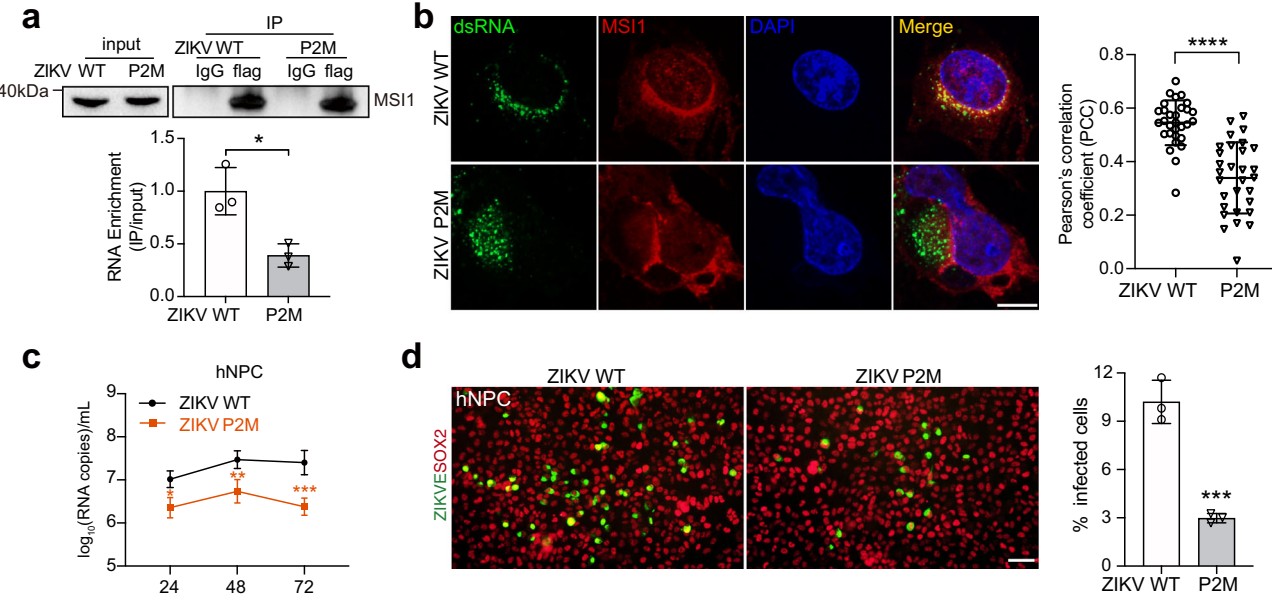

**Fig. 4 | The xrRNA2 AGAA tetraloop participates in MSI1 binding in vivo and regulates ZIKV replication. a** RNA-IP analysis from WT or P2M infected BHK-21 cells transfected with flag-MSI1. Western blot shows immunoprecipitations (IPs) by immunoglobulin G (IgG) and flag antibodies. Input (5%) represents whole-cell extract. Western blot was probed with antibodies against flag. The bound viral RNA from the IP was analyzed by qRT-PCR. Graph below shows qRT-PCR performed on bound RNA from IP. RNA-IP values are presented as the ratio to the input after subtraction of the IgG background. Data are mean ± SD. $n = 3$ independent experiments. Two-sided Student's $t$ test, *$P < 0.05$. ($P = 0.0237$). **b** Confocal microscopy of BHK-21 cells transduced with MSI1-expressing lentiviruses (MOI = 10) and infected with the ZIKV WT or P2M viruses (MOI = 1) for 24 h. Fixed cells were immunostained with anti-dsRNA and anti-MSI1 antibodies. Cell nuclei were counterstained with DAPI. Scale bar, 10 μM. The graph on the right shows Pearson's correlation coefficient (PCC), and mean ± SD was calculated from 30 cells in each group. Two-sided Student's $t$ test. ****$P < 0.0001$. ($P < 0.0001$). **c** hNPCs were infected with the ZIKV WT or P2M viruses (MOI = 1), and the culture supernatants were harvested at the indicated time points for detection of viral RNA copies by qRT-PCR. Data are mean ± SD. $n = 3$ independent experiments. Two-way ANOVA, *$P < 0.05$, **$P < 0.01$, ***$P < 0.001$. (24 h $P = 0.0139$; 48 h $P = 0.0069$; 72 h $P = 0.0005$). **d** The expression of viral envelope protein at 48 h after infection from (c) was detected by immunostaining. Scale bar, 50 μm. mean ± SD. Two-sided Student's $t$ test, ***$P < 0.001$. ($P = 0.0008$). Source data are provided as a Source Data file.

fractions equal to or above 60% after 5 minutes incubation, suggesting RRM12 has a highly dynamic conformation in solution (Fig. S7b). Both RRM1 and RRM2 adopt a βαββαβ fold topology[16,19,35,36]. Four peptides corresponding to residues 35-42 (α₁) and 76-85 (α₂) of RRM1 and residues 107–120 (β₁) and 142-157 (β₃) of RRM2 showed substantial decrease in HDX kinetics in the *holo* state (Fig. S7b, c), implying that they may be at or close to the binding interfaces (Fig. 7b). Canonical RRM-RNA binding interactions indicate that conserved residues in the β₁ and β₃ strands of RRMs are involved in single-stranded RNA recognition, while residues at helices α₁ and α₂ are not[16,19,35,36]. The HDX-MS data indicate that RRM2 uses the canonical RNA recognition interfaces whilst RRM1 may utilize a non-canonical interface.

To validate the predicted RRM1-RNA interface, we monitored the effect of alanine substitutions of residues residing in the non-canonical interface identified by HDX-MS (α₁: R37A/F40A/F43A, α₂: K76A/R82A) or the canonical RNA recognition interface (β₁: K21A/F23A, β₃: F63A/F65A) in RRM1 by ITC experiments. SAXS data demonstrated that the described mutations didn't cause significant structural changes to RRM12 (Fig. S7e–g and Table S1). While substitutions of residues in the canonical interface (β₁ and β₃) of RRM1 did not impair xrRNA2 binding, substitutions of residues in α₁ and α₂ attenuated xrRNA2 binding (Fig. S7h, Table S2). Thus, RRM1 utilizes a non-canonical interface at or close to helices α₁ and α₂ to bind xrRNA2.

### Structural investigation of the MSI1-xrRNA2 complex by integrative methods

To investigate the potential binding model of MSI1 in complex with xrRNA2, an integrative structural approach that combines SAXS, interface information from mutagenesis and HDX-MS experiments with computational modeling was utilized[37]. We first characterized the solution structure of the MSI1 RRM12 in complex with xrRNA2 at low resolution by SAXS (Fig. 7). The overall 3D shape envelopes for RRM12, xrRNA2 and the RRM12-xrRNA2 complex were ab initio reconstructed from the SAXS data[38] (Fig. 7a–e). Both the atomic models of MSI1 and xrRNA2 by SAXS-driven homology modeling can be nicely fitted into the corresponding 3D shape envelopes by DAMMIN (Fig. 7b, c). The two-phase modeling of the RRM12-xrRNA2 complex by MONSA[38] suggests a side-by-side interaction between the protein and RNA (Fig. 7d).

We initially performed de novo docking of RRM12 and xrRNA2 complex without experimental restraints using HADDOCK2.4 (high ambiguity driven docking) platform[39,40]. The generated models fit SAXS data badly (χ² greater than 10) and the binding interfaces are conflicting with experimental data, such as RRM12 binding to P4 or PK2 of xrRNA2. We next performed information-driven flexible docking on the RRM12-xrRNA2 complex. The binding interface restraints from mutagenesis and HDX-MS data as well as the radius of gyration (R_g) restraint from SAXS data were supplemented during integrative modeling by HADDOCK 2.4. HADDOCK clustered 230 structures in 10 clusters, with Z-scores ranging from −1.4 to 1.3 (the smaller the better). The experimental SAXS scattering profile and the 3D shape envelope by DAMMIN were used to validate the models. As shown in Fig. 7e–g, the representative models from the top 2 best clusters (Z-scores of −1.4 and −1.2, respectively) fit well with the experimental SAXS data (fitting χ² of 2.5 and 3.2, respectively) and the shape envelope by DAMMIN. The binding interfaces between xrRNA2 and RRM12 in these models are also compatible with the mutagenesis and HDX-MS data. The models from clusters 3–10 either fit the SAXS data poorly (fitting χ² larger than 10), or are not compatible with the binding interface data (Fig. S8), are therefore not further discussed. The representative

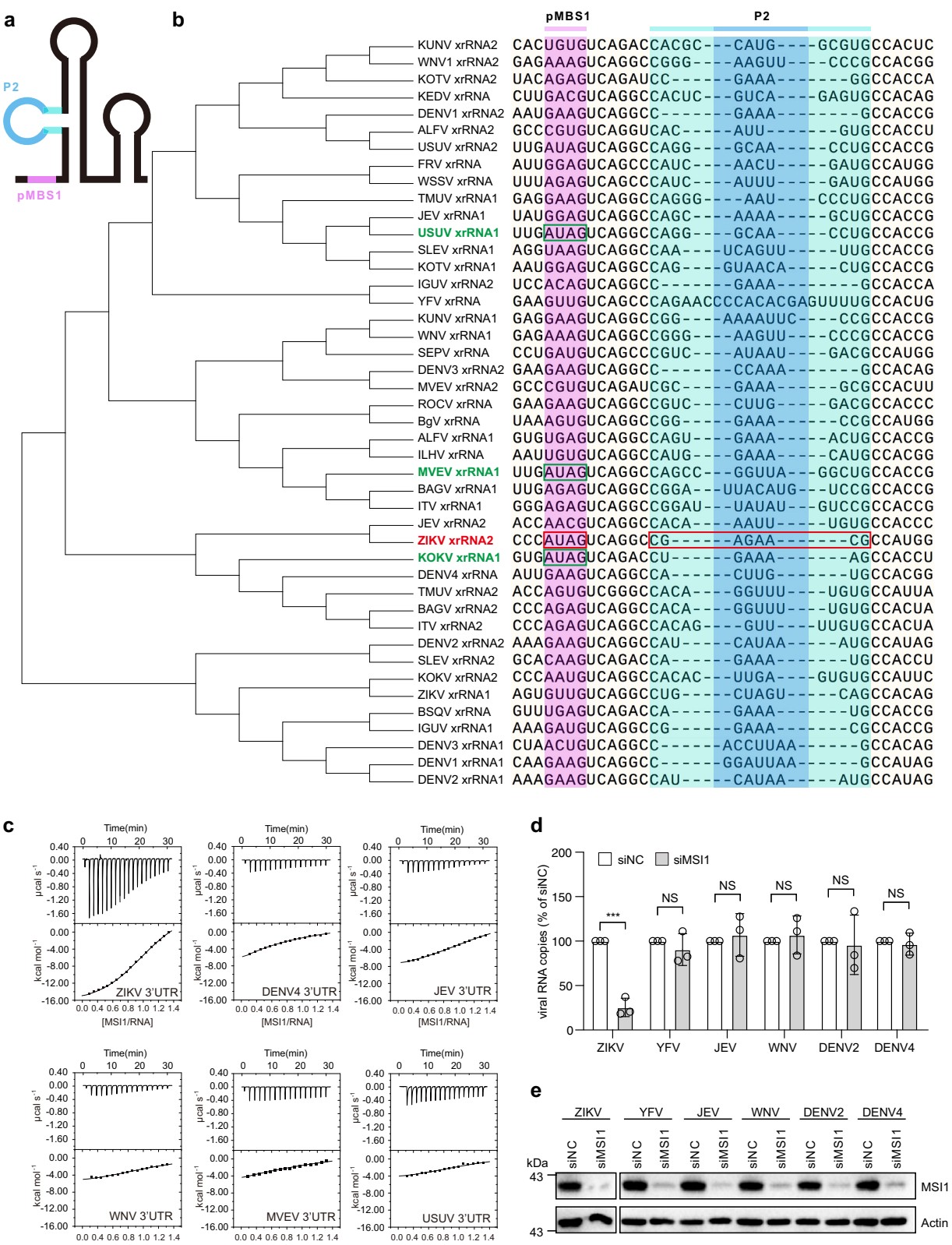

model from cluster 1(Z-score of −1.64), as shown in Fig. 7f, reveals a bipartite binding mode, which the pMBS1 and P2 AGNN tetraloop of xrRNA2 act like a pincer to specifically target the noncanonical RNA binding interface in RRM1 (α1 and α2) and loop3 connecting the β2 and β3 strands in RRM2 of MSI1, respectively. These binding interfaces between xrRNA2 and RRM12 are also compatible with the HDX-MS and mutagenesis data. In the representative model from cluster (Z-score of

1.2), RRM2 is assumed to utilize the canonical RNA binding interfaces of the β1 (RNP2) and β3 (RNP1) strands as well as loop3 (connecting β2 and β3) to recognize the pMBS1 (U3A4G5), and RRM1 recognizes the AGNN-type P2 of xrRNA2 through its α-helices (Fig. 7g). The interaction of RRM1 with the AGAA tetraloop in P2 is reminiscent of the recognition of the AGNN tetraloop by the double-stranded RNA binding domain (dsRBD) of Rnt1p RNase III33, where the dsRBD

**Fig. 5 | The two MBSs of xrRNA2 are unique among MBFV. a** Secondary structure diagram of flaviviral xrRNA with the pMBS1 colored in pink, the P2 loop colored in blue and the P2 stem colored in cyan. **b** A maximum likelihood tree constructed with the xrRNA sequences of flaviviruses using MEGA. The pMBS1 and P2 loop sequences (colored as in (**a**)) of xrRNAs are shown after the virus names. The pMBS1 and P2 loop sequences of ZIKV xrRNA2 are framed with red rectangles. The pMBS1 sequences of the xrRNA1s of KOKV, USUV, and MVEV, which match AUAG are framed with green rectangles. The GenBank ID information of sequences used for this analysis were listed in Table S4. **c** ITC profiles of RRM12 to JEV/YFV/DENV2/

DENV4/WNV 3´ UTRs. **d** Relative viral RNA copies in the supernatants of control siRNA (siNC) and MSI1 siRNA (siMSI1) treated hNPCs after infection with different flaviviruses (ZIKV: MOI = 1; YFV: MOI = 1; JEV: MOI = 1; WNV: MOI = 0.1; DENV2: MOI = 1; DENV4: MOI = 1). **e** Representative Western blots of hNPCs treated with control and MSI1 siRNAs. Blots were probed with antibodies against MSI1 or actin as a loading control. Data were expressed as the mean ± SD. $n = 3$ independent experiments. Two-sided Student's $t$ test, \*\*\*$P < 0.0001$, NS not significant. (ZIKV $P = 0.0002$). Source data are provided as a Source Data file.

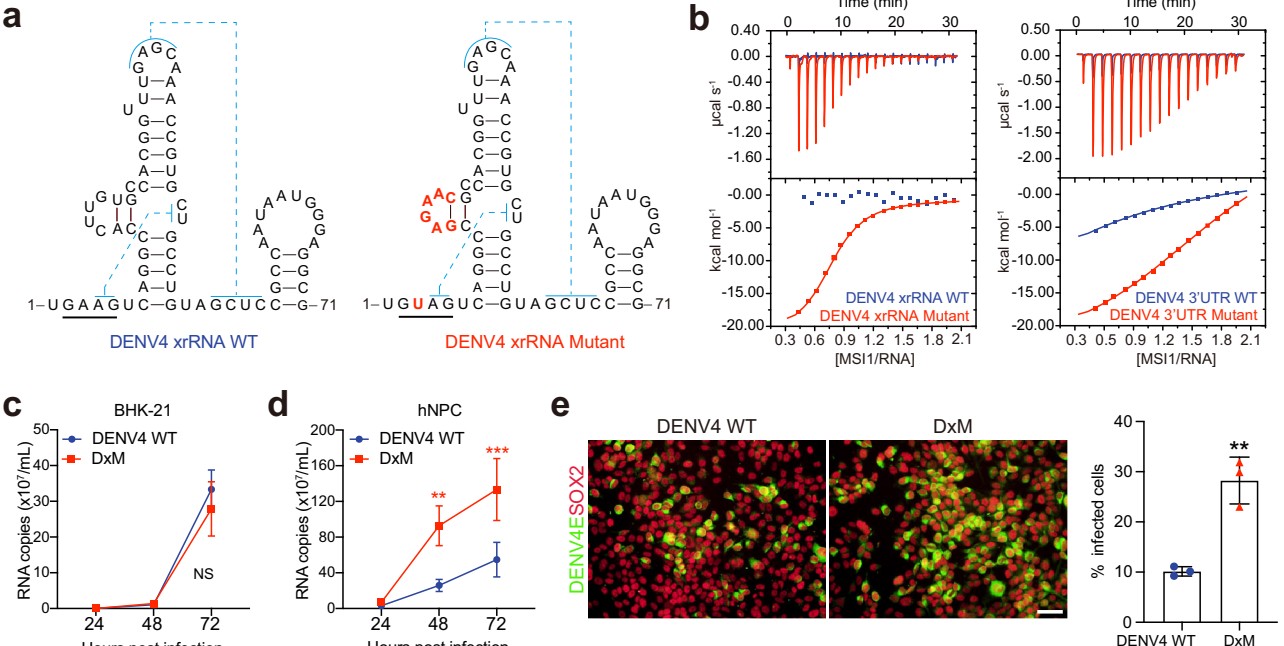

**Fig. 6 | The MSI1 binding element of ZIKV xrRNA2 could confer MSI1 binding ability to DENV4. a** Secondary structure of DENV4 xrRNA and the xrRNA mutant. **b** ITC profiles of RRM12 to DENV4 xrRNA and DENV4 xrRNA mutant (left); ITC profiles of RRM12 to DENV4 3´UTR and DENV4 3´UTR mutant (right). **c** BHK-21 cells were infected with the DENV4 WT or DxM viruses (MOI = 0.1), and the culture supernatants were harvested at the indicated time points for the detection of viral RNA copies by qRT-PCR. Data are mean ± SD. $n = 3$ independent experiments. Two-way ANOVA, NS not significant. **d** hNPCs were infected with the DENV4 WT or DxM

virus (MOI = 1), and the culture supernatants were harvested at the indicated time points for detection of viral RNA copies by qRT-PCR. Data are mean ± SD. $n = 3$ independent experiments. Two-way ANOVA, \*\*$P < 0.01$, \*\*\*$P < 0.001$. (48 h $P = 0.0028$; 72 h $P = 0.0008$). **e** The expression of viral envelope protein at 48 h after infection from (**d**) was detected by immunostaining. Scale bar, 50 μm. mean ± SD. $n = 3$ independent experiments. Two-sided Student's $t$ test, \*\*$P < 0.01$. ($P = 0.0027$). Source data are provided as a Source Data file.

contacts the RNA at successive minor, major, and AGNN tetraloop minor grooves on one face of the helix. Taken together, the models in cluster 2 also indicated a bipartite binding mode between MSI1 and xrRNA2.

## Discussion

Here, we demonstrated ZIKV utilizes its canonical pMBS1 (5´-AUAG-3´) and a novel non-canonical MBS, the AGNN-type tetraloop in xrRNA2 to specifically target the RRM1 and RRM2 subdomains of MSI1 in a bipartite mode. Of the three pMBSs in the ZIKV 3´-UTR, only pMBS1 in xrRNA2 (5´-AUAG-3´) is critical for MSI1 binding and viral replication in hNPCs. Thus, the AUAG motif represents the authentic canonical MBS, which is well documented in host RNA transcripts. Surprisingly, we also revealed that the AGAA tetraloop of xrRNA2 is also critical for MSI1 binding and the unique cell tropism of ZIKV, thus representing a second but non-canonical MBS. More importantly, the MSI binding affinity is completely dependent on the cooperation of the two MBSs.

Our findings have significant implications. First, the identification of the AGNN-type tetraloop as an MBS changed the canonical rule of MSI1 binding targets. All previous RNA targets of MSI1 are identified as

an A/GU(1-3)AG motif[11,13,18,21,22], this long-standing information may be incomplete. Our findings contribute to a better understanding of the complexity of MSI1 binding. Whether similar structures are present in host RNA transcripts remains to be determined. This unexpected binding model of ZIKV xrRNA2 with MSI1 suggests the known MSI1 targets of host RNA transcripts, such as numb[11] and p21[20], may be more complex and include unidentified RNA structures. Second, our findings provide a long-sought answer to the question of why is only ZIKV linked to neurodevelopmental diseases. A wide variety of viruses possess the canonical pMBS sequence in their 3´ UTRs[26] (Fig. S5, Fig. 5a), but none of them has been proven to have the ability to bind MSI1. The unique MBS of ZIKV thus provides new evidence supporting the causal link between ZIKV infection and the distinctive congenital malformation in infants. Finally, our results highlight a new role of xrRNA2. The only known function of the highly structured xrRNA in flaviviruses was to block the degradative activity of host exoribonuclease Xrn-1 resulting in the generation of sfRNAs[32,41], which facilitate viral replication, by multiple actions including blocking the host IFN response[42,43], suppressing RNAi[44,45] and apoptosis[46]. Our findings suggest that, in addition to producing sfRNAs, ZIKV xrRNA2 can also use

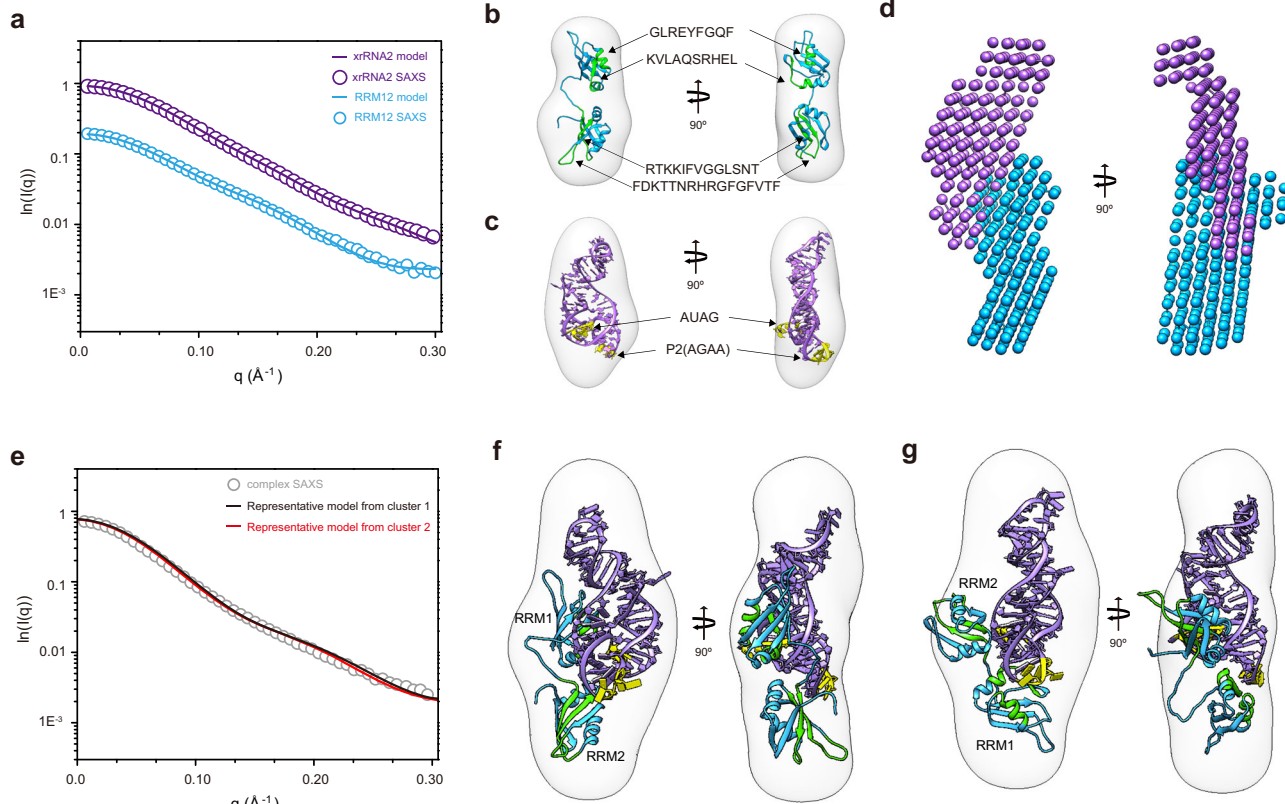

**Fig. 7 | Structural investigation of the MSI1 RRM12-xrRNA2 complex by integrative methods. a** Structural characterization of free RRM12 and free xrRNA2 by SAXS. The back-calculated scattering profiles (solid line) of the best models were overlaid with the respective experimental profiles (open cycle). **b** The homology model of RRM12 (blue) was further subjected to rigid body refinement against SAXS data by Xplor-NIH and fitted into the envelope ab initio. The residues presumably involved in xrRNA2 binding identified by HDX-MS are highlighted in green. **c** The best homology model of xrRNA2 (purple) was fitted into the envelope ab initio. The pMBS1 and AGAA tetraloop of xrRNA2 are highlighted in yellow. **d** The 3D shape envelope of RRM12-xrRNA2 complex reconstructed ab initio with the multiphase modeling program MONSA, and the bead models for RRM12 and xrRNA2 are colored according to (**b**) and (**c**). **e** Structural characterization of RRM12-xrRNA2 complex by SAXS. The back-calculated scattering profiles (solid line) of the representative models from cluster 1 (black) and cluster 2 (red) are overlaid with the respective experimental profile (open cycle). **f, g** The representative model from cluster 1 (**f**) and cluster 2 (**g**) are fitted into the 3D shape envelope ab initio reconstructed by DAMMIN. RRM12 and xrRNA2 are colored as in (**b**) and (**c**).

its unique RNA motif and tetraloop fold to target MSI1 which determines viral replication in specific cells, like NPCs.

The structural study of RNA-protein complexes using conventional high-resolution techniques such as X-ray crystallography (XRC), cryo-EM and NMR remains challenging, and currently only about 2.2% of the structures in the Protein Data Bank represents protein-RNA complexes (PDB)[47]. Using an integrative approach combining SAXS, HDX-MS, mutagenesis, and computational modeling, we presented two potential bipartite binding models for hMSI1 RRM12 and ZIKV xrRNA2, which precisely explain the results obtained from biochemical and virological assays. Given that MSI1 utilizes its two RRMs to interact with pMBS1 and the P2 stem-loop of xrRNA2, two possible binding modes of RRM12 with xrRNA2 can be speculated: 1) The pMBS1 of xrRNA2 binds to RRM2 and the AGNN-type P2 stem of xrRNA2 binds to RRM1 (mode 1); 2) The pMBS1 of xrRNA2 binds to RRM1 and the AGNN-type P2 stem binds to RRM2 (mode 2). Our HDX-MS data indicated that RRM1 interacts with xrRNA2 with a noncanonical interface (α1 and α2), while RRM2 interacts with xrRNA2 with the conserved canonical interface (β1 and β2, as well as loop3 that connects β1 and β2). The NMR structures of mouse MSI1 RRM1 and RRM2 in complex with single-stranded RNA (AUAG, pMBS1) have shown that both RRMs utilize the canonical RNA binding motif to interact with single-stranded RNA[16,19]. The interaction of AGNN tetraloop with α-helix in proteins has also been reported[33,48]. Thus the binding mode 1 in models from cluster 2 contains more RNA-protein interaction features similar to that in the

published structural data. However, it's currently difficult to ascertain which binding mode is accurate due to the low resolution of SAXS data. A high-resolution structure of the MSI1-xrRNA2 complex should be pursued in the future.

Overall, our study not only identifies an unusual viral RNA element that contributes to the unique tropism of ZIKV, but also illustrates the precise mechanism and structural basis by which a virus utilizes a primary RNA sequence motif and a tertiary RNA structure to target a specific host protein (Fig. S9).

## Methods

### Ethics statement

All virus studies were performed in strict accordance with the guidelines set by the Academy of Military Medical Sciences (AMMS). Approval to use JEV vaccine strain SA14-14-2 was obtained from the Chengdu Institute of Biological Products. Approval to use YFV vaccine strain 17D was obtained from the Beijing Institute of Biological Products. Informed consent was obtained during the isolation of ZIKV and DENV strains.

### Cells and viruses

The baby hamster kidney fibroblast (BHK-21) cells (ATCC CCL10), African green monkey kidney epithelial (Vero) cells (ATCC CCL81) were cultured in DMEM (Thermo Fisher Scientific) containing 7% fetal bovine serum (FBS, Biowest). C6/36 cells (ATCC CRL-1660) were

cultured in PRMI 1640 (Thermo Fisher Scientific) containing 10% FBS (Biowest). Human Neuroblastoma cell line SH-SY5Y (ATCC CRL-2266) was cultured using DMEM/F12 (Thermo Fisher Scientific) containing 10% FBS. Human neural progenitor cell (hNPC) line 15167 derived from fetal brains (Lonza) was kindly provided by S. Bao (Cleveland Clinic). hNPC and U251 were cultured as neurospheres in neurocult-XF basal medium (STEMCELL technologies) supplemented with neurocult-XF proliferation supplement (STEMCELL technologies), basic fibroblast growth factor (bFGF, 10 ng/mL, STEMCELL technologies), and epidermal growth factor (EGF, 20 ng/mL, STEMCELL technologies), Heparin solution (2ug/mL, STEMCELL technologies). For the BHK-21-Msi1 cells stably expressing MSI1, the MSI1 encoding sequence (NM_002442.4) was constructed into lentiviral vector LV-6, containing a puromycin resistance gene by Genepharma (Shanghai, China). BHK-21 cells were infected with lentivirus with an MOI of 100. Forty-eight hours after infection, cells were screened with 1 ug/mL puromycin for a week. MSI1 expression was detected by IFA and Western Blot. BHK-21-Ctrl cells were constructed by infection with LV-6 lentiviral empty vector.

ZIKV strain FSS13025 (GenBank accession number KU955593) was originally isolated from a patient in Cambodia in 2010[31]. The JEV vaccine strain SA14-14-2 and YFV vaccine strain 17D were from the Chengdu Institute of Biological Products and Beijing Institute of Biological Products, respectively. WNV strain NY99 was rescued from the two-plasmid infectious clone[49]. DENV2 strain 43 (GenBank accession number AF204178.1) was isolated from dengue fever (DF) patient in China in 1987. DENV4 strain B5 (GenBank accession number AF289029.1) was isolated from a DF patient in China[50].

## Antibodies
MSI-1(rabbit, ab52865, Abcam), SOX2(rabbit, ab97959, Abcam), Flag (mouse-F1804, sigma), dsRNA-J2 (mouse, 1010200, Scicons), ZIKV envelope protein (mouse, BF-1176-46, Biofront technologies), FMRP (rabbit, A4539, abclonal), DENV envelope protein (2A10, mouse, produced in our lab), Actin (rabbit, AC026, Abclonal), mouse IgG isotype control (ab37355, Abcam), goat anti-rabbit IgG (alexa fluor 594) (ab150080, Abcam), goat anti-mouse IgG (alexa fluor 488) (ab150113, Abcam), horseradish peroxidase-conjugated goat anti-rabbit IgG (ZB-2301, ZSGB-BIO), horseradish peroxidase-conjugated goat anti-rabbit IgG (ZB-2305, ZSGB-BIO).

## Protein expression and purification
DNA encoding human MSI1 RRM12 (20-191), RRM1 (20-103), and RRM2 (109-191) were amplified from the human Musashi-1 gene by polymerase chain reaction (PCR) and subcloned into pGEX-6p-1. MSI1 RRM12 cysteine mutants (C49/C167/C181S (for double labeling), C49/C167S/C181S (for single labeling) and C49S/C167/C181S (for single labeling)) for EPR experiment were generated by the Quick-change site-directed mutagenesis strategy and verified by DNA sequencing. The information on primers used was provided in Table S3. The proteins were expressed with an N-terminal GST tag followed by a PreScission protease cleavage site in *Escherichia coli* Rosetta (DE3) at 16 °C in LB medium. Cells were grown to $OD_{600}$-0.8 and induced with 0.5 mM IPTG for 14 h, then harvested by centrifugation and resuspended with lysis buffer (20 mM HEPES pH 7.5, 2 M NaCl, 2 mM DTT and 2 mM EDTA). Cells were further disrupted by high pressure homogenizer and centrifuged at 18,000 $g$ for 1 h. The soluble fraction containing fusion proteins was purified by GST-affinity chromatography (GE healthcare). After washed with washing buffer (20 mM HEPES pH 7.5, 2 M NaCl, 2 mM DTT), PreScission protease was supplemented into the column with a mass ratio of 1:100 and incubated at room temperature for 4 h to remove GST tag. The fractions containing target proteins were collected and diluted with buffer A (20 mM HEPES pH 7.5, 20 mM NaCl, 2 mM DTT), and loaded into Hitrap Heparin

column (GE healthcare), followed with gradient elution with buffer B (20 mM HEPES pH 7.5, 1 M NaCl, 2 mM DTT). The target fractions were collected and stored at −80 °C for further use.

## RNA transcription and purification
The respective RNAs were prepared by in vitro transcription and purified as previously reported[51]. Briefly, plasmids encoding an upstream T7 promoter and the respective RNAs were total-gene synthesized and confirmed by DNA sequencing (Wuxi Qinglan Biotechnology Inc, Wuxi, China). Using these plasmids as templates, plasmids encoding the respective RNA mutants or subdomains were generated by the Quick-change site-directed mutagenesis strategy and verified by DNA sequencing. The information of primers used was provided in Table S3. The double-stranded DNA fragment templates for in vitro RNA production were generated by PCR using an upstream forward primer targeted the plasmids and a downstream reverse primer specific to respective cDNAs. The RNAs were transcribed in vitro using T7 RNA polymerase and purified by preparative, non-denaturing polyacrylamide gel electrophoresis.

## Isothermal titration calorimetry (ITC)
ITC experiments were performed at 25 °C on a MicroCal PEAQ-ITC instrument (Malvern Instruments). Proteins and RNAs were buffer exchanged into 20 mM HEPES pH 7.5, 150 mM NaCl, 3 mM MgCl$_2$ with Superdex 75 size exclusion column (GE healthcare). The sample cell containing 300 μL of 30 μM RNA was titrated with 17 successive injections of 600 μM protein. Acquired titration curves were fitted with the Origin 7.0 program using the "one set of binding sites" biding model.

## flag-msi1 plasmid construction
The cDNA of human MSI1 (NM_002442.4) was amplified from the RNA of hNPCs with the primers: 5' GACGACGATGACAAGGGATCCATGGA GACTGACGCGCC 3' and 5' AAGCTTCTGCAGGTCGACTCAGTGGTA CCCATTGG 3', and cloned into pRK5-Flag vector by Gibson Assembly Cloning Kit (E5510S, NEB).

## Generation of mutant viruses
The substitutions construction and viral RNA transcription were performed as described above[52]. The primers used for site-directed mutagenesis for ZIKV- pMBS1M were: forward 5' ACCAAGCCCAa AGTCAGGCCG 3'; reverse 5'TTCCCAGCTTCTCCTGGG 3', for ZIKV-P2M were: forward 5'TAGTCAGGCCCgaAAgGCCATGGCACG 3'; reverse 5'TGGGCTTGGTTTCCCAGC 3', for ZIKV-AGCU were: forward 5' TCA GGCCGAGctCGCCATGGCA 3'; reverse 5' CTATGGGCTTGGTTTCCC 3' and for DENV4-DxM were: forward 5' CCgagaacGCCACGGTTTG AGCAAAC 3'; reverse 5' CCTGACTaCAATAGCCTTTGGTGTTTGTTG 3'. The RNA was then transfected into BHK-21 cells using Lipofectamine 3000 reagent (Thermo Fisher Scientific). Culture supernatants were collected at 72 h post-transfection and used to infect C6/36 cells maintained in RPMI 1640 (Thermo Fisher Scientific) containing 2% FBS (Biowest). The C6/36 cells were cultured at 28 °C for 5−7 days and then the supernatants were dispensed into single-use aliquots, which were stored at −80 °C. Infectious virions were detected by plaque assay and viral antigen expression was detected by indirect immunofluorescent assay. The titers of virus stocks were then determined by plaque-forming assay, and the substitution sites were confirmed by RT-PCR and DNA sequencing.

## Plaque assay
Virus samples were serial-diluted by 10-fold with DMEM containing 2% FBS and 400 μl of each dilutes were added onto vero cells in 12-wells and incubated at 37 °C under 5% CO2 for 2 h. The supernatants were then aspired and 1 mL of DMEM containing 1% low melting point

agarose (Promega) and 2% FBS was added into each well. Four days post-infection, cells were fixed with 4% formaldehyde for 2 h at room temperature, followed by staining with 1% crystal violet solution for 30 min. After rinsing with water, the number of Plaques were counted for the calculations of virus titers.

## RNA pulldown

Mutated ZIKV 3'UTR DNA were synthesized and inserted into vector PUC57-KAN by Sangon (Shanghai, China) and then mutated ZIKV 3'UTR DNA fragment containing T7 promoters were amplified from them by PCR. The wild type ZIKV 3'UTR DNA fragment containing T7 promoter was amplified from the infectious clone. Then biotinylated WT and mutated ZIKV 3'UTR RNAs were synthesized by in vitro transcription using RiboMAX™ Large Scale RNA Production Systems-T7 (Promega) according to the manufacturer's instructions with the modification of the components of rNTPs were changed to: 8 mM GTP, 5 mM ATP, 5 mM CTP, 1.3 mM UTP, 0.7 mM Bio-11-UTP (Thermo Fisher Scientific). synthesized RNAs were purified using Purelink RNA mini kit (Thermo Fisher Scientific) and checked by agarose electrophoresis. BHK-21 cells were transfected with flag-msi1, forty-eight hours after transfection, cells were lysed by a lysis buffer (50 mM Tris-HCl, pH 8.0, 150 mM NaCl, 1 % NP-40, 5 mM EDTA, 10 % glycerol, 10 mM dithiothreitol, 1X cocktail protease inhibitor (YTHX Biotechnology) and 100 U/mL recombinant RNASe inhibitor (Takara)). For the pulldown, 300ug of total protein from the cell lysates was precleared with 30uL Dynabeads MyOne Streptavidin T1 beads (Invitrogen). Biotinylated RNA was previously incubated at 60 °C for 5 min and then slowly cooled to room temperature. The precleared cell lysate was incubated with 5pmol RNA at 4 °C for 3 h, Subsequently, 30 uL Dynabeads MyOne Streptavidin T1 beads were added for another 2 h. After incubation, the beads were washed five times with lysis buffer. After washing, the beads were resuspended in 50 μL of 2X SDS loading buffer and boiled for 10 min at 98 °C. The samples were centrifuged and the supernatants were analyzed by Western Blot.

## RNA immunoprecipitation (RIP)

BHK-21 cells were transfected with flag-msi1 plasmids. Twenty-four hours after transfection, cells were infected with ZIKV. Forty-eight hours after infection, cells were lysed by a lysis buffer (50 mM Tris-HCl, pH 8.0, 150 mM NaCl, 1 % NP-40, 5 mM EDTA, 10 % glycerol, 10 mM dithiothreitol, 1X cocktail protease inhibitor (YTHX Biotechnology) and 100 U/mL recombinant RNASe inhibitor (Takara)). The cell lysates were subjected to immunoprecipitation with 0.5 ug anti-FLAG antibody or IgG control at 4 °C overnight and then mixed with 20 μl of gammabind G sepharose (GE Healthcare) and incubated for another 3 h. Beads were washed five times with lysis buffer and resuspended in 100 uL of lysis buffer. 50 uL of beads were used for Western Blot and 50 uL for RNA isolation using RNeasy Mini Kit (Qiagen). qRT-PCR was performed by use of the 3'UTR specific primers that amplify ZIKV 3'UTR (nucleotides 10623 to 10722 of ZIKV-Cambodia: forward 5'CCTGAACTGGAGATCAGCTGTG 3'; reverse 5' GGTCTTTCCCAGCG TCAATA 3').

## Replicon assay

The ZIKV replicon that carries the Renilla Luciferase gene was previously described[53]. The substitutions were constructed using the Q5 site-directed mutagenesis kit (NEB). Primers used for site-directed mutagenesis were as the same as primers used for the generation of mutant viruses described above. The replicon plasmids were linearized by restriction endonuclease digestion and purified by Phenol/Chloroform extraction. In vitro transcribed viral RNA was prepared using Ribomax T7 large scale RNA production kit (Promega) and purified using Purelink RNA mini kit (Thermo Fisher Scientific). $2 \times 10^4$ cells

were seeded into each well of 48-well plates and incubated at 37 °C in 5% CO2. One day after seeding, 500 ng of each replicon RNA was transfected into each well of cells using Lipofectamine 3000 reagent (Thermo Fisher Scientific). The cell lysates were collected at the given time points and the luciferase activity assay was performed using the Renilla luciferase assay system (Promega) in a GloMax Discover system (Promega).

## Immunofluorescence analysis

Cells were fixed with 1% paraformaldehyde for 15 min and permeabilized with 0.2% Triton X-100 for 15 min, blocked with 3% bovine serum albumin for 2 h at 37 °C, and then incubated in the corresponding primary antibody (MSI1, 1:1000; sox2, 1:1000; ZIKV E, 1:1000), and then washed with PBS (3 × 5 min), followed by incubating in the secondary antibody (1:1000) at 37 °C for 1 h. After three washes with PBS, the cell nuclei were stained with DAPI (Merck D9542). Cells were photographed under an Olympus IX73 microscope. For the co-localization analysis, the samples were analyzed using a Nikon Eclipse Ti-U confocal microscope. Co-localization was calculated by Pearson's correlation coefficient (PCC) using FIJI with the Coloc2 plugin[54].

## Western blot

The samples were fractionated by electrophoresis on 10% SDS-polyacrylamide gels, and resolved proteins were transferred onto PVDF membranes. After blocking with 5 % skimmed milk, the membranes were incubated with in the corresponding primary antibody (MSI1, 1:1500; FMRP, 1:1000; Actin, 1:10000; ZIKV E, 1:1000), and then washed with 0.05% tween-20 in PBS (4 × 5 min), followed by appropriate horseradish peroxidase-conjugated secondary antibodies (1:10000), and then washed with 0.05% tween-20 in PBS (4 × 5 min). Blots were developed using an enhanced chemiluminescence (ECL) kit (Thermo Fisher Scientific).

## Growth curve analysis

BHK-21 and SH-SY5Y cells were seeded onto 24-well plates and cultured overnight. For hNPC and U251, wells were previously coated with Matrigel Matrix (354248, corning) for 3 h at 37 °C with 5% CO2, the Matrigel Matrix then was discarded and cells were seeded. Cells were infected at indicted MOI. Infected cells were cultured at 37 °C with 5% CO2 and culture supernatants were collected and the cells were fixed at the indicated time points. Viral RNA was extracted using Purelink RNA minikit (Thermo Fisher Scientific) and analyzed by qRT-PCR. qRT-PCR was performed using One Step PrimeScript™ RT-PCR kit (Takara) with the primers: forward 5' GGTCAGCGTCCTCTCTAATAAACG3'; reverse 5' GCACCCTAGTGTCCACTTTTTCC 3' and probe: FAM-AGCC ATGACCGACACCACACCGT-BQ1.

## RNA interference

A total of 30 pmol MSI1siRNA (sc-106836 santa cruz) or control siRNA (sc-37007 santa cruz) per well was transfected in a 24-well plate using RNAiMAX regeant (Thermo Fisher Scientific) as recommended by the manufacturer. The medium was changed at 6 h after transfection. Twenty-forty hours after transfection, cells were infected with viruses at indicted MOI. Forty-eighty hours after infection, cells supernatant and lysates were harvested for analysis.

## SAXS sample preparation, data collection, and processing

All the RNAs and proteins were buffer exchanged into SAXS buffer (20 mM HEPES pH 7.5, 150 mM NaCl, 3 mM MgCl2, 5 mM DTT and 5% v/v glycerol) with size exclusion chromatography. To prepare RRM12-xrRNA2 complex, ZIKV xrRNA2 and MSI1 RRM12 were incubated at room temperature with a molar ratio at 1:3, and further purified through Superdex 75 size exclusion column with SAXS buffer. All the samples were concentrated to ~3 mg/ml for SAXS experiment.

Concentration series measurements (4- and 2-fold dilution and stock solution) were carried out and no aggregation or repulsion were observed. SAXS data was collected at room temperature at the beamline 12 ID-B of the Advanced Photon Source, Argonne National Laboratory as previously described[51]. Briefly, the setup was adjusted to achieve scattering $q$ values of $0.05 < q < 0.089$ Å$^{-1}$, where $q = (4\pi/\lambda) \sin\theta$, and $2\theta$ is the scattering angle. Thirty 2D images for each buffer or sample were recorded using a flow cell with the exposure time of 1 second. No radiation damage was observed as confirmed by the absence of systematic signal changes in sequentially collected X-ray scattering images. The 2D images were reduced to one-dimensional scattering profiles by MATLAB onsite. The scattering profile, the forward scattering intensity I(0) and the radius of gyration ($R_g$), the pair distance distribution function (PDDF) as well as the maximum dimension, $D_{max}$ of the molecules were calculated using the same procedures as described before. The Volume-of-correlation ($V_c$) was calculated by using the program Scatter, and the molecular weights of molecules were calculated on a relative scale using the $R_g/V_c$ power law developed by Rambo et al.[55]. The theoretical scattering intensities of the atomic structure models were calculated and fitted to the experimental scattering profile using CRYSOL[56].

### ab initio shape reconstruction and all-atom 3D atomic modeling

Low-resolution bead models of MSI1 RRM12, xrRNA2, and their complex were built up with the program DAMMIN, which generates models represented by an ensemble of densely packed beads, using SAXS scattering data within the $q$ range of $0.006–0.3$ Å$^{-1}$. Forty independent runs were performed, and the resulting models were subjected to averaging by DAMAVER and were superimposed by SUPCOMB based on the normalized spatial discrepancy (NSD)[38].

The atomic model of human MSI1 RRM12 was built up with MODELLER using the NMR structure of Mouse RRM1 (PDB ID: 2RS2) and RRM2 (PDB ID: 5X3Z) as templates[16,19]. The atomic model of MSI1 RRM12 was further refined against SAXS data with Xplor-NIH, which RRM1 (20–96) and RRM2 (109–191) were kept as rigid bodies to translate and rotate and the linker between (97–108) was allowed to translate or rotate freely. The atomic model of ZIKA xrRNA2 was built up with the FARFAR2, using the crystal structure of ZIKA xrRNA1 (PDB ID: 5TPY) along with NMR structure of AGAA tetraloop (PDB ID: 1K4A) as templates[32,57].

The multiphase ab initio modeling program MONSA was used to obtain dummy atom model of RRM12-xrRNA2 complex. Twenty independent runs gave reproducible models, and the typical model is calculated using DAMAVER suite[38].

### Hydrogen-deuterium exchange mass spectrometry (HDX-MS)

For HDX-MS experiment, 20 µL of 5 mg/ml MSI1 RRM12 alone or in the presence of xrRNA2 were prepared. The pH of the buffer was 7.6, which contained 20 mM HEPES, 100 mM KCl, 3 mM MgCl$_2$, and 0.5 mM TCEP. To initiate deuterium labeling, 5 µl of each 5 mg/ml protein solution was diluted with 45 µL of labeling buffer (contents, 99% D$_2$O, pH 7.2) at room temperature for 60 s, 90 s, and 300 s. The reaction was quenched with 50 µL of ice-cold quench buffer (1% (v/v) formic acid in water solution, 100% H$_2$O). The reaction tube was then put on ice. 10 µL of 1 µM pepsin solution was added for digestion. After 5 min, the sample was centrifuged and placed into the auto-sampler of the Ultimate 3000 UPLC system (Thermo, CA, USA) for injection. The column and samples were precooled on ice. Fifty microlitres of sample was then loaded onto and separated by a ACQUITY UPLC 1.7 µM BEHC18 1.0 × 50 mm column (Waters). The peptides were eluted by a 16-min gradient of acetonitrile (1 to 50%) in 0.1% formic acid at 100 µL/min with Thermo-Dionex Ultimate 3000 HPLC system. Mass spectrometry analysis was performed on Q Exactive Orbitrap mass spectrometer (Thermo, CA). The peptides were identified by using an in-house Proteome Discover (version PD1.4, Thermo Fisher Scientific), and HDX-

MS data were processed by HDExaminer from Thermo Fisher Scientific. The peptides were assigned as high or medium confidence by using HDExaminer (Sierra Analytics). The confidence level is calculated using a number of factors, including signal to noise and how well the theoretical isotope cluster matches the actual data[58]. The deuterium percentages were computed with fully deuterated samples. The fully deuterated samples were prepared by incubating sample with deuterium-exchange solution at room temperature for 24 h, and then quench the reaction with 36 µL of optimal quench solution. The deuterium uptake was determined by monitoring shifts of the centroid peptide isotopic distribution by using the program HDExaminer (Thermo Fisher Scientific). The hydrogen/deuterium exchange difference of each peptide between protein alone and protein with ligand was manually checked. The experiments were repeated three times to ensure repeatability and deliver an estimate of the precision in the measurement.

### Integrative modeling of MSI1 RRM12-xrRNA2 complex

The structure of RRM12-xrRNA2 complex was predicted using the integrative modeling platform HADDOCK2.4 package[39,40]. The individual RRM1 (aa: 20-96) and RRM2 (aa: 109-190) domains rather than the integral RRM12 molecule were docked on xrRNA2, which allows large amplitude motions between RRMs to occur during docking. The restraints used to guide sampling and evaluate the docked poses include: (1) The scattering profile and radius of gyration ($R_g$) of the RRM12-xrRNA2 complex by SAXS; (2) The putative interaction interfaces between RRM12 and xrRNA2 derived from HDX-MS and mutagenesis data, and prior structural information on MSI1 RRM-ssRNA$^{GUAGU}$ complex (PDB ID: 2RS2 and 5X3Z). To facilitate extensive sampling in the conformational space of the RRM12-xrRNA2 complex, we increase the number of structures at the initial stage from 1000 (default value) to 10,000, and increase the steps of sampling at stages of docking (Table S6). From the xrRNA2 atomic model, the nucleobase of G$^5$ within pMBS1 forms a canonical Watson-crick base pair with C$^{42}$ in PK1. The importance of PK1 to the fold and functional integrity of flaviviral xrRNAs have been described before[59]. G$^5$ might be embedded in the core of xrRNA2 3D ring-like structure and solvent inaccessible to RRM12 (Fig. S10A-C), thus only the U$^3$A$^4$ dinucleotides within pMBS1 are likely to interact with RRM12 during docking. The nucleotides 1–4 of xrRNA2 are allowed to be flexible during refinement.

Following the spirit of Ambiguous Interaction Restraints exploited by HADDOCK[40], we defined the residues which are high solvent accessible and evidenced to be involved in binding as active residues and set the solvent-accessible neighbors of active residues as passive residues. The detailed information regarding the active and passive residues for each interface can be found in Table S7.

The SAXS scattering profile and the 3D shape envelope of MSI1 RRM12-xrRNA2 complex were used to score the representative models from the respective clusters generated by HADDOCK. The program CRYSOL was employed to calculate the theoretical scattering profile of the respective models and fit with the experimental scattering profile[56].

### Structural illustration

The bead models generated by DAMMIN were transformed into volumetric maps using Situs[60]. All of the illustrations of atomic models were generated using the Chimera[61] or PyMOL Molecular Graphics System, version 1.3, Schrödinger, LLC.

### Statistical analysis

Sample size was estimated on the basis of similar research reported in the literature and no statistical method was used to predetermine sample size. No data were excluded from analyzes. All experiments were performed using at least 3 biological replicates to ensure

reproducibility. All samples were analyzed equally with no sub-sampling. Therefore, there was no requirement for randomization. Investigators were generally not blinded as the experimental conditions required investigators to know the identity of the samples. Description of the statistical tests performed for each experiment and the *p*-values can be found in the corresponding figure legends. Dots on the bar graphs represent the values from an individual replicate of the experiment. All data were analyzed using Prism software (GraphPad). For RNA enrichment, PCC, and virus infection rate data, unpaired two-sided Student's *t* test was performed to estimate the statistical significance between two groups. For virus growth curve analysis, data were analyzed by two-way ANOVA with Bonferroni correction for multiple comparisons. Representative data from repeated independent experiments are presented as mean ± standard deviation (SD) with triplicate samples. Detailed statistical results can be found in the Source Data file.

### Reporting summary

Further information on research design is available in the Nature Portfolio Reporting Summary linked to this article.

## Data availability

All data are available with the article, Supplementary Information or Source Data file. Source data are provided as a Source Data file. Source data are provided with this paper.

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

## Acknowledgements

We thank Dr. Andrew Davidson from the University of Bristol for kind suggestion and discussion, Drs. Xiaolin Tian and Haiteng Deng in Center of Protein Analysis Technology, Tsinghua University for HDX-MS analysis, Cuiyan Zhou in Protein Preparation and Identification Facility, Tsinghua University for assistance with ITC experiments, and the staff at the beamline 12-ID-B, Argonne National Laboratory, USA, and beamline BL19U2, Shanghai Synchrotron Radiation Facility, China, for assistance during SAXS data collection. This work was supported in part by the National Key Research and Development Project of China (2018YFA0900801 (H.Z.), 2021YFC2302400 (C.-F.Q.), and 2021YFA1301500 (X.F.)), the National Natural Science Foundation (NSFC) (32100127 (R.L.), U1832215 (X.F.), and 31770190 (C.-F.Q.)), the Postdoctoral Science Foundation of China (2019M664017) (X.C.), C.-F.Q. was supported by the National Science Fund for Distinguished Young Scholars (81925025), the Innovative Research Group (81621005) from the NSFC, and the Innovation Fund for Medical Sciences (2019-I2M-5-049) from the Chinese Academy of Medical Sciences.

## Author contributions

C.F.Q. and X.F. conceived, designed and supervised the study. X.C., Y.W. and Z.X. performed the majority of the experiments and analyzed the data. X.Z. performed the SAXS measurements. Q.M. and Z.W. conducted immunostaining. M.C. and R.L. performed the sequence alignment and analysis under the guidance of H.Z. X.L. edited the draft. All authors read and approved the contents of the manuscript.

## Competing interests

C.F.Q. and X.F. have filed a patent related to the major findings reported in this study. The remaining authors declare no competing interests.
