## [Peer Review File · Nature Communications]

Zika virus RNA structure controls its unique neurotropism by bipartite binding to Musashi-1REVIEWER COMMENTS

Reviewer #1 (Remarks to the Author):

The study of Xiang Chen and co-authors focuses on identifying the molecular mechanism of Musashi-1 binding to the Zika virus (ZIKV) 3'-untranslated region (3'UTRs). Musashi-1 has been previously shown to interact with the ZIKV 3'UTR to enhance viral replication in neural cells and to serve as an important host factor for viral neurovirulence. Musashi binding sites (MBSs) were found in the genomes of multiple RNA viruses, however it remained unexplained why Musashi-RNA interaction only occurs in ZIKV. The study of Chem et al provides the explanation for this phenomenon. Using combination of biophysical and molecular methods authors established that MBS alone is not sufficient for effective Musashi binding and requires additional binding site in the tetraloop region within the viral xrRNA2 structure. Authors validated the mechanism and functional significance of this interaction using mutagenesis approach and identified the structure of xrRNA-Musashi complex. They also demonstrated that although MBSs are present in multiple flaviviruses, the tetraloop required for Musashi binding is only present in ZIKV, which explains why this interaction is unique to this virus. Generally, this is a well-executed study which answers a long-standing research question and will definitely be of interest for virologists and RNA biologists.

I have the following concerns that need to be addressed:

1. In Fig 4 authors show that mutations that destroys AGNN tetraloop and remove Musashi binding in vitro result in attenuated ZIKV phenotype in human NPC. Authors should also make viruses with mutations that preserve AGNN tetraloop which showed no effect Musashi binding in vitro also have no effect on virus replication.
2. Authors should assess by Northern blot whether mutation in xrRNA2 tetraloop affects production of sfRNA, as it can also be a contributing factor to the attenuated virus phenotype shown in Fig.4.

Suggestion:

Authors use qRT-PCR for viral RNA to assess viral replication (Fig. 1F, 2A,E, 4C, etc). The use of this method is sub-optimal as viral RNA in culture fluids does not necessarily reflect production of infectious particles (virus replication). It is affected by RNA leakage from dying cells and presence of defective infectious particles. It would be more convincing if the authors could validate growth kinetics results using one of the virological assays such as plaque assay, foci-forming assay or TCID50.

Reviewer #2 (Remarks to the Author):

Chen et al. present a manuscript where they study a role of MSI1 binding to Zika virus (ZIKV) 3'UTR. The binding of MSI1 to ZIKV 3' UTR has been previously reported. Since MSI1 is expressed in the neuronal progenitor cells this was linked to the ZIKV tropism towards developing brains and resulting pathophysiology. As the viral tropism towards neuronal cells is still poorly understood this are of research is of great significance.

In this study, authors found that ZIKV has several canonical MSI1 binding sites (MBS) and through biochemical as well as life virus studies they managed to prove that one MBS (pMBS1) in the xrRNA2 serves is a major binding platform. The binding of MSI1 to pMBS1 has positive effect on virus replication in the presence of MSI1 and ZIKV with mutated pMBS1 are not sensitive to the MSI1 expression. Moreover, through careful structural and evolutionary analyses authors found that there is another non-canonical MSI1 binding site, which contributes to a bipartite binding of MSI1 to ZIKV 3'UTR. I find this manuscript interesting and novel but I do have a number of major concerns:

1. A pull down experiment (Fig 1E) is missing a control in the form of some other, unrelated RBP, which is not affected by the mutation in the pMBS1. This is crucial to prove that the pMBS1m RNA is intact and equally stable as compared with the WT RNA in the course of the experiment.
2. Where is the evidence that MSI1 is not expressed in the BHK-21 cells? Also, western blots showing the comparative levels in all tested cell lines (with and without knock down and over-

expression) should be presented in the manuscript.

3. The RNA-IP assay in Fig. 1G does not show decreased immunoprecipitation of MSI1 by the pMBS1m RNA compared with the ZIKV WT RNA. If anything, looking at the reduced levels of MSI1 in the input and similar signal of MSI1 in RNA-IP, my conclusion is that the pMBS1m RNA binds MSI1 better than the WT in cell. Similar comment can be made about the RNA-IP data presented in Fig 4. Additionally, controls of unrelated RBP, which will not be affected by the mutations in the 3'UTR are missing.

4. A transplant of the ZIKV MSI1 bipartite binding motive into other viruses should be attempted. If these mutated viruses acquire neural progenitor tropism and are sensitised to the levels of MSI1 it would greatly increase the impact of this manuscript.

Minor point:

1. The total length of the ZIKV 3'UTR should be mentioned in the text and indicated in the Fig. 1A.

Reviewer #3 (Remarks to the Author):

In this manuscript, the authors provide a well-written and robust study that characterizes the sequence-determinants of MSI1 binding to the 3'UTR of ZIKV. They discover a novel interaction site in the RNA with MSI1, and determine that this site is unique to ZIKV among flaviviruses. The authors demonstrate that this interaction is critical for RNA-MSI1 interactions *in vitro* and *in vivo* and that this interaction is pro-viral in MSI1-expressing cells, which importantly includes neural progenitor cells (hNPCs). Overall this is a very well-executed study providing novel insights into the mechanisms of pathogenesis of ZIKV.

I have no major concerns or comments.

An additional piece of evidence that would support the authors' claims that this interaction is responsible (ultimately) for hNPC tropism/cytopathology of ZIKV, would be to introduce the novel MSI1 binding site into another flavivirus and demonstrate that this also enhances the replication of this virus in hNPCs (or MSI1-expressing BHK cells). (e.g. adding the AGAA tetraloop into KOKV, MVEV, or USUV, or adding both the AUAG and AGAA sites into e.g. DENV4). Of course, this would likely constitute a gain-of-function experiment. Therefore, I wonder if the authors can use a replicon system for one of these other viruses or demonstrate a gained interaction between an *in vitro* expressed 3'UTR and MSI1 upon introduction of the ZIKV MSI1 binding sites?

Reviewer #4 (Remarks to the Author):

The manuscript by Chen et al., investigates the interaction between MSI1 and its binding site in the ZIKV RNA. For the most part, the experiments are well described and are a good effort to narrow down and identify the binding interactions involved in this protein-RNA interaction. The models generated look reasonable, but more discussion is needed to validate how the sum of the results supports the proposed models. There are a few major concerns regarding the SAXS and HDX-MS data that will need to be addressed prior to publication.

1) The authors claim their model is consistent with the *ab initio* shapes – but those are really low resolution – anything might fit inside there (by eye). The authors should attempt to make alternative structures with the 'wrong' interface and test whether there is actually a better fit to the SAXS data with their 'correct' structure. The more appropriate way to do this would be using Crysol or FoXS to test the Chi2 among many potential models.

At the end of the results the manuscript presents two models from the HADDOCK clusters, but it is unclear which the authors think is more correct. The statistics on the chi2 values are reported, but is one of the clusters more consistent with the data, beyond SAXS? I think the authors can add a few more sentences here to discuss which cluster is more consistent with all of the available data.

In addition to the two clusters presented in figure 6, were there others? If so, then those should be discussed too. It would be really helpful to see how another 'good' potential model would look when compared to the ab initio bead models and the overall SAXS profile (χ^2). I think seeing this would add more confidence to the analysis that the SAXS data really is sufficient for restraining potential models.

2) In the description of the SAXS data collection, the authors say the data was recorded with 1 second exposures through a flow cell. Is this sufficient to prevent the possibility of radiation damage? I like that the buffer included DTT and glycerol, as these will help minimize radiation damage, but was there another control in place to account for potential radiation damage? Often, researchers will compare the first exposures to the last ones to ensure nothing in the sample was damaged during the data collection.

Also, how do the authors account for potential oligomerization or repulsion of the samples at the high concentrations used for SAXS? Was there an additional control at lower concentrations to ensure the low angle SAXS data are consistent? The authors should consult the following paper: <https://onlinelibrary.wiley.com/doi/10.1002/pro.351>

and make sure they account for any concentration effects and radiation damage that might mislead the modeling efforts.

3) The HDX-MS is utilized for the major conclusions of the paper, but the data presented and the description of the analysis is lacking. The only real data shown for HDX-MS is in figure S6, and even that doesn't include anything that can be used to assess the quality of the data. How was the deuterium uptake analyzed? How were peptides identified? The authors only state that it was manually checked. Were there any replicates? How do they assign statistically relevant changes? The authors need to include two supplementary files: 1) a summary of the exchange; and 2) the corresponding experimental statistics parameters used for the analysis, as outlined in the general recommendations paper from 2019: <https://www.nature.com/articles/s41592-019-0459-y>

4) Other notes on the HDX-MS portion of the paper:

The sequence coverage should include the actual sequence of the protein, and not just the amino acid numbers.

How were peptides assigned as high or medium confidence?

The differential uptake plots should show all the relevant timepoints and either error bars or lines to indicate significance thresholds. The text argues that the apo form was highly dynamic, but none of the figures actually show how dynamic the protein is overall. You don't get this information from the differential uptake plot.

The authors use an LC-MS system that apparently does not cool the sample or the column. Is this actually the case? Was the sample chamber in the autosampler or the LC column and lines cooled? The authors also say they used a 37 minute gradient, which is exceedingly long for HDX-MS, especially without any cooling of the column. If this is the case, then it is impossible to see 80% deuterium retention as some of the figures indicate. At 293K and pH 2.5 you would, on average, be losing 75% of all amide deuterium within 12 minutes. What was the actual level of back-exchange in the HDX-MS experiments?

How were the percentages in figure S6C computed? Was there a fully deuterated control?

The mutagenesis helps alleviate some of this concern, but as the HDX was the basis of those mutations, and is still used to guide modeling, all of the associated methods and analysis of the HDX is something that really needs to be clarified and explained.

5) Minor issue: I'm guessing the column used for HDX-MS was a 1x50mm (millimeter, not microMolar).

Reviewer's Comments:

Reviewer #1 (Remarks to the Author)

The study of Xiang Chen and co-authors focuses on identifying the molecular mechanism of Musashi-1 binding to the Zika virus (ZIKV) 3'-untranslated region (3'UTRs). Musashi-1 has been previously shown to interact with the ZIKV 3'UTR to enhance viral replication in neural cells and to serve as an important host factor for viral neurovirulence. Musashi binding sites (MBSs) were found in the genomes of multiple RNA viruses, however it remained unexplained why Musashi-RNA interaction only occurs in ZIKV. The study of Chem et al provides the explanation for this phenomenon. Using combination of biophysical and molecular methods authors established that MBS alone is not sufficient for effective Musashi binding and requires additional binding site in the tetraloop region within the viral xrRNA2 structure. Authors validated the mechanism and functional significance of this interaction using mutagenesis approach and identified the structure of xrRNA-Musashi complex.

They also demonstrated that although MBSs are present in multiple flaviviruses, the tetraloop required for Musashi binding is only present in ZIKV, which explains why this interaction is unique to this virus. Generally, this is a well-executed study which answers a long-standing research question and will definitely be of interest for virologists and RNA biologists.

Response: Thanks for the encouraging comments.

I have the following concerns that need to be addressed:

1. In Fig 4 authors show that mutations that destroys AGNN tetraloop and remove Musashi binding in vitro result in attenuated ZIKV phenotype in human NPC. Authors should also make viruses with mutations that preserve AGNN tetraloop which showed no effect Musashi binding in vitro also have no effect on virus replication.

Response: Thanks for the insightful comments. We have produced a mutant ZIKV, P2-AGCU, in which the xrRNA2 P2 was replaced with the sequence 5' CG-AGCU-CG 3', keeping AGNN-type of P2 unchanged. The P2-AGCU mutation did not affect MSI1

binding affinity (Fig. 3E). We compared the replication kinetics of WT ZIKV and P2-AGCU ZIKV in BHK-21 cells and hNPCs, and the results showed that the P2-AGCU ZIKV and WT ZIKV replicated equally well in BHK-21 cells and hNPCs. This new result has been included in the revised manuscript (**Fig. S4**).

2. Authors should assess by Northern blot whether mutation in xrRNA2 tetraloop affects production of sfRNA, as it can also be a contributing factor to the attenuated virus phenotype shown in Fig.4.

Response: Thanks for the helpful comments. We have detected the sfRNA production of WT ZIKV and mutant ZIKVs by Northern blot, and the result showed that the pMBS1m mutant and the xrRNA2 P2 mutations did not affect sfRNA production. This new result has been included in new **Fig. S2C** and **Fig. S3J** in the revised manuscript.

Suggestion:

Authors use qRT-PCR for viral RNA to assess viral replication (Fig. 1F, 2A,E, 4C, etc). The use of this method is sub-optimal as viral RNA in culture fluids does not necessarily reflect production of infectious particles (virus replication). It is affected by RNA leakage from dying cells and presence of defective infectious particles. It would be more convincing if the authors could validate growth kinetics results using one of the virological assays such as plaque assay, foci-forming assay or TCID50.

Response: Thanks for the suggestion. We agree with the reviewer that it would further benefit our study to validate growth kinetics results using one of the assays such as plaque assay, foci-forming assay or TCID50. However, most samples are extracted as RNAs during our initial experiments. Whatever, we compared growth curves obtained from plaque assay and qRT-PCR methods of ZIKV on BHK-21 cells and hNPCs. As shown in response Fig.1, the virus titers of infectious ZIKV were well correlated with the copy numbers of ZIKV RNA, and a strong linear correlation was observed between them. Thus, our qRT-PCR experimental system could be used to assess the viral replication.

Response Fig. 1. (A). Comparison of the growth curves obtained from plaque assay and qRT-PCR methods of ZIKV on BHK-21 and hNPCs. (B). Regression analysis between viral titers and RNA copy numbers.

Reviewer #2 (Remarks to the Author)

Chen et al. present a manuscript where they study a role of MSI1 binding to Zika virus (ZIKV) 3'UTR. The binding of MSI1 to ZIKV 3' UTR has been previously reported. Since MSI1 is expressed in the neuronal progenitor cells this was linked to the ZIKV tropism towards developing brains and resulting pathophysiology. As the viral tropism towards neuronal cells is still poorly understood this area of research is of great significance.

In this study, authors found that ZIKV has several canonical MSI1 binding sites (MBS) and through biochemical as well as life virus studies they managed to prove that one MBS

(pMBS1) in the xrRNA2 serves as a major binding platform. The binding of MSI1 to pMBS1 has a positive effect on virus replication in the presence of MSI1 and ZIKV with mutated pMBS1 are not sensitive to the MSI1 expression. Moreover, through careful structural and evolutionary analyses authors found that there is another non-canonical MSI1 binding site, which contributes to a bipartite binding of MSI1 to ZIKV 3'UTR. I find this manuscript interesting and novel but I do have a number of major concerns:

Response: Thanks for the encouraging comments.

1. A pull down experiment (Fig 1E) is missing a control in the form of some other, unrelated RBP, which is not affected by the mutation in the pMBS1. This is crucial to prove that the pMBS1m RNA is intact and equally stable as compared with the WT RNA in the course of the experiment.

Response: Thanks for the helpful comments. According to the reviewer's suggestion, we have detected the binding of WT 3'UTR, pMBS1m, pMBS2 and pMBS3m with another unrelated RBP, FMRP, which was previously reported to bind to ZIKV 3'UTR (R. Soto-Acosta *et al*, *Elife*, 2018). The result showed that the pMBS mutations did not affect the interaction between 3'UTR and FMRP. The new result was included in new **Fig. 1E**.

2. Where is the evidence that MSI1 is not expressed in the BHK-21 cells? Also, western blots showing the comparative levels in all tested cell lines (with and without knock down and over-expression) should be presented in the manuscript.

Response: Thanks for the helpful comments. We have added the western blots results of the MSI1 expression in hNPC, SH-SY5Y, U251, BHK-21, BHK-21-ctrl and BHK-21-MSI1 cells in new **Fig. S2E**. The result showed that MSI1 is highly expressed in hNPC, SH-SY5Y and U251 cells but not detected in BHK-21 cells. Meanwhile, MSI1 is well expressed in MSI1-overexpression cell line BHK-21-MSI1. The MSI1 knock down result was presented in Fig. 2D and Fig. 5E.

3. The RNA-IP assay in Fig. 1G does not show decreased immunoprecipitation of MSI1 by the pMBS1m RNA compared with the ZIKV WT RNA. If anything, looking at the reduced levels of MSI1 in the input and similar signal of MSI1 in RNA-IP, my conclusion is that the pMBS1m RNA binds MSI1 better than the WT in cell. Similar comment can be made about the RNA-IP data presented in Fig 4. Additionally, controls of unrelated RBP, which will not be affected by the mutations in the 3'UTR are missing.

Response: The reviewer might misunderstand the results shown in Fig. 1G and Fig.4. In the RNA-IP assay, the MSI1 protein was immunoprecipitated by antibody, and the RNAs interacting with MSI1 were also isolated and analyzed by RT-qPCR. In Fig. 1G, the western blots result in upper panel represents the input MSI1 and the efficiency of MSI1 immunoprecipitation. Thus, the similar levels of MSI1 indicated the same immunoprecipitation efficiencies. The graph below shows the levels of bound viral RNA. The reduced bound viral RNA level in pMBS1m infection indicated the pMBS1m mutation reduced the binding between MSI1 and viral RNA.

We thank the reviewer's suggestion on control RBP. Accordingly, we performed additional RNA-IP experiments to detect the binding between FMRP (the control RBP) and ZIKV RNA. The result showed WT ZIKV RNA and pMBS1m ZIKV RNA bind equally well to FMRP, and the similar result was observed in P2M mutation. These new results were included in new **Fig. S2D** and **Fig. S3I**.

4. A transplant of the ZIKV MSI1 bipartite binding motive into other viruses should be attempted. If these mutated viruses acquire neural progenitor tropism and are sensitised to the levels of MSI1 it would greatly increase the impact of this manuscript.

Response: Thanks for the constructive comments. We have transplanted the ZIKV MSI1 bipartite binding element into DENV4 xrRNA, generating a DENV4 xrRNA mutant and a DENV4 3'UTR mutant. ITC results indicated that this transplantation can confer DENV4 xrRNA and 3'UTR high MSI1 binding affinity. We also introduced this mutation into a DENV4 infectious clone, and constructed a mutant DENV4 (DxM). Then the virus replication in hNPCs was analyzed. The result showed that DxM

exhibited enhanced replication efficiency in hNPCs. These data confirmed the important role of ZIKV MSI1 binding element in viral replication in hNPCs. This new result has been included in the revised manuscript (**Fig. 6**).

Minor point:

1. The total length of the ZKIV 3'UTR should be mentioned in the text and indicated in the Fig. 1A.

Response: Thanks for the comments. We have added the total length information of the ZKIV 3'UTR in the text (**line 58**) and Fig. 1A in the revised manuscript.

Reviewer #3 (Remarks to the Author)

In this manuscript, the authors provide a well-written and robust study that characterizes the sequence-determinants of MSI1 binding to the 3'UTR of ZIKV. They discover a novel interaction site in the RNA with MSI1, and determine that this site is unique to ZIKV among flaviviruses. The authors demonstrate that this interaction is critical for RNA-MSI1 interactions in vitro and in vivo and that this interaction is pro-viral in MSI1-expressing cells, which importantly includes neural progenitor cells (hNPCs). Overall this is a very well-executed study providing novel insights into the mechanisms of pathogenesis of ZIKV.

Response: Thanks for the encouraging comments.

I have no major concerns or comments.

An additional piece of evidence that would support the authors claims that this interaction is responsible (ultimately) for hNPC tropism/cytopathology of ZIKV, would be to introduce the novel MSI1 binding site into another flavivirus and demonstrate that this also enhances the replication of this virus in hNPCs (or MSI1-expressing BHK cells). (e.g. adding the AGAA tetraloop into KOKV, MVEV, or USUV, or adding both the AUAG and AGAA sites

into e.g. DENV4). Of course, this would likely constitute a gain-of-function experiment. Therefore, I wonder if the authors can use a replicon system for one of these other viruses or demonstrate a gained interaction between an in vitro expressed 3'UTR and MSI1 upon introduction of the ZIKV MSI1 binding sites?

Response: Thanks for the insightful comments. We have transplanted the ZIKV MSI1 bipartite binding element into DENV4 xrRNA, generating a DENV4 xrRNA mutant and a DENV4 3'UTR mutant. ITC results indicated that this transplantation can confer DENV4 xrRNA and 3'UTR high MSI1 binding affinity. We also introduced this mutation into a DENV4 infectious clone, and constructed a mutant DENV4 (DxM). Then the virus replication in hNPCs was analyzed. The result showed that DxM exhibited enhanced replication efficiency in hNPCs. These data confirmed the important role of ZIKV MSI1 binding element in viral replication in hNPCs. This new result has been included in the revised manuscript (**Fig. 6**).

Reviewer #4 (Remarks to the Author):

The manuscript by Chen et al., investigate the interaction between MSI1 and its binding site in the ZIKV RNA. For the most part, the experiments are well described and are a good effort to narrow down and identify the binding interactions involved in this protein-RNA interaction. The models generated look reasonable, but more discussion is needed to validate how the sum of the results supports the proposed models. There are a few major concerns regarding the SAXS and HDX-MS data that will need to be addressed prior to publication.

1) The authors claim their model is consistent with the ab initio shapes – but those are really low resolution – anything might fit inside there (by eye). The authors should attempt to make alternative structures with the 'wrong' interface and test whether there is actually a better fit to the SAXS data with their 'correct' structure. The more appropriate way to do this would be using Crysol or FoXS to test the Chi2 among many potential models.

Response: We agree with the reviewer's comments. Our initial *de novo* docking of

RRM12-xrRNA2 complex without experimental restraints failed to generate convergent models and the resulting models fit SAXS data badly. Among these models, some RRM12 bind to P4 or PK2 in xrRNA2, which interfaces are not consistent with the RNA mutagenesis data. This is the reason why we perform HDX-MS experiment to obtain protein interface information and then utilize the integrative modeling strategy to improve the docking performance. We validate the resulting models from the 10 clusters using SAXS scattering profile (using Crysol) and 3D shape envelope, and found the models from the top 2 best clusters are consistent with the experimental data. The HADDOCK z-scores of all the clusters, and the fitting χ^2 of the respective representative models was reported in the revised manuscript (**Fig. 7** and **Fig. S8**).

At the end of the results the manuscript presents two models from the HADDOCK clusters, but it is unclear which the authors think is more correct. The statistics on the chi2 values are reported, but is one of the clusters more consistent with the data, beyond SAXS? I think the authors can add a few more sentences here to discuss which cluster is more consistent with all of the available data.

Response: Both the representative models from the top 2 best clusters are consistent with the SAXS data, the protein and RNA interfaces from mutagenesis and HDX-MS data and reveal a bipartite binding mode. Due to the low resolution of SAXS data, it is hard to say which model is more accurate. From the literature, the NMR structures of mouse mushshi-1 RRM1 and RRM2 in complex with single stranded RNA (AUAG, pMBS1) has been reported, both RRMs utilize the canonical RNA binding motif to interact with single-stranded RNA. The recognition of AGAA tetraloop by α -helix of RNase III has also been reported. From our HDX-MS data, RRM1 interact with xrRNA2 with a noncanonical α -helix interface ($\alpha 1$ and $\alpha 2$), while RRM2 interact with xrRNA2 with the conserved interface ($\beta 1$ and $\beta 2$, as well as loop3 that connect $\beta 1$ and $\beta 2$). Thus, the RNA-protein binding mode in the representative model from cluster 2 is more compatible with the published high-resolution structures. We discussed this in the revised manuscript, please see **line 378-391**.

In addition to the two clusters presented in figure 6, were there others? If so, then those should be discussed too. It would be really helpful to see how another 'good' potential

model would look when compared to the ab initio bead models and the overall SAXS profile (χ^2). I think seeing this would add more confidence to the analysis that the SAXS data really is sufficient for restraining potential models.

Response: We followed the suggestions from reviewer 4 and showed the models from cluster 3-10 in Fig. S8 as a comparison. The HADDOCK Z-scores and fitting χ^2 was included in Fig. S8 as well. We discussed these clusters in the revised manuscript: “The models from clusters 3-10 either fit the SAXS data poorly (fitting χ^2 larger than 10), or are not compatible with the binding interface data (Fig. S8), are therefore not further discussed.” Please see **line 324-326** in the revised manuscript.

2) In the description of the SAXS data collection, the authors say the data was recorded with 1 second exposures through a flow cell. Is this sufficient to prevent the possibility of radiation damage? I like that the buffer included DTT and glycerol, as these will help minimize radiation damage, but was there another control in place to account for potential radiation damage? Often, researchers will compare the first exposures to the last ones to ensure nothing in the sample was damaged during the data collection.

Also, how do the authors account for potential oligomerization or repulsion of the samples at the high concentrations used for SAXS? Was there an additional control at lower concentrations to ensure the low angle SAXS data are consistent? The authors should consult the following paper:

<https://onlinelibrary.wiley.com/doi/10.1002/pro.351>

and make sure they account for any concentration effects and radiation damage that might mislead the modeling efforts.

Response: The parameters for SAXS data collection described in the manuscript is sufficient to prevent radiation damage. During SAXS data collection, we collected thirty 2D images for each sample and the 1st one-dimensional scattering profile overlaps very well with the 30th one-dimensional scattering profile, as well as other scattering profiles, indicated that no radiation damage is observed (see Response figure 2A, using RRM12-xrRNA2 sample as an example). We supplemented “No radiation damage was observed as confirmed by the absence of systematic signal changes in sequentially

collected X-ray scattering images” to the Material and method in the revised manuscript to underline that no radiation damage was observed.

Despite the high-concentration sample, we also collected SAXS data with sample at two-fold and four-fold diluted concentration. For RRM12-xrRNA2 complex, the concentration is 3 mg/mL, 1.5 mg/mL and 0.75 mg/mL for each sample, respectively. The scattering profile of each concentration sample overlays well with each other, and the liner guinier fitting also indicated that samples are monodispersed instead of aggregated or repulsion (see Response Figure 2B). We also supplemented this detail to the material and method in the revised manuscript: “Concentration series measurements (4- and 2-fold dilution and stock solution) were carried out and no aggregations or repulsion were observed.”

Response Figure 2. (A) The overlap of 30 scattering profiles of RRM12-xrRNA2. (B) The scattering profile of RRM12-xrRNA2 at different concentration.

3) The HDX-MS is utilized for the major conclusions of the paper, but the data presented and the description of the analysis is lacking. The only real data shown for HDX-MS is in figure S6, and even that doesn't include anything that can be used to assess the quality of the data. How was the deuterium uptake analyzed? How were peptides identified? The authors only state that it was manually checked. Were there any replicates? How do they assign statistically relevant changes? The authors need to include two supplementary files: 1) a summary of the exchange; and 2) the corresponding experimental statistics parameters used for the analysis, as outlined in the general recommendations paper from 2019: <https://www.nature.com/articles/s41592-019-0459-y>

Response: The deuterium uptake was determined by monitoring shifts of the centroid peptide isotopic distribution by using the program HDExaminer (Thermo Fisher Scientific). The peptides were identified by using an in-house Proteome Discoverer (version PD1.4, Thermo Fisher Scientific), and HDX-MS data were processed by HDExaminer from Thermo Fisher Scientific. The experiments were repeated 3 times to ensure repeatability and deliver an estimate of the precision in the measurement. We added the above details in the revised manuscript, please see the revised supplementary information.

4) Other notes on the HDX-MS portion of the paper:

The sequence coverage should include the actual sequence of the protein, and not just the amino acid numbers.

Response: We added the protein sequence in the sequence coverage figure, please see **Fig. S7A** in the revised manuscript.

How were peptides assigned as high or medium confidence?

Response: The peptides were assigned as high or medium confidence by using HDExaminer (Thermo Fisher Scientific). The confidence level is calculated using a number of factors, including signal to noise and how well the theoretical isotope cluster matches the actual data. We supplemented above information in the revised manuscript.

The differential uptake plots should show all the relevant timepoints and either error bars or lines to indicate significance thresholds. The text argues that the apo form was highly dynamic, but none of the figures actually show how dynamic the protein is overall. You don't get this information from the differential uptake plot.

Response: Three timepoints for recording were at 60 s, 90 s, and 300 s, which has already shown in Fig. S7C. We also supplemented the differential plots of deuterium uptake of peptides at all the timepoints in the revised **Fig. S7D**. We added the error bars as reviewer 4 suggested, please see **Fig. S7D** in the revised manuscript. We also supplemented the raw deuterium uptake plots of RRM12 alone and in the presence of

xrRNA2 in the revised manuscript. The high deuterium uptake fractions of *apo* form RRM12 could support our conclusion that RRM12 alone is dynamic in solution.

The authors use an LC-MS system that apparently does not cool the sample or the column. Is this actually the case? Was the sample chamber in the autosampler or the LC column and lines cooled? The authors also say they used a 37 minute gradient, which is exceedingly long for HDX-MS, especially without any cooling of the column. If this is the case, then it is impossible to see 80% deuterium retention as some of the figures indicate. At 293K and pH 2.5 you would, on average, be losing 75% of all amide deuterium within 12 minutes. What was the actual level of back-exchange in the HDX-MS experiments?

Response: The column and samples were precooled on ice, and the peptides were eluted by a 16-min gradient of acetonitrile (1 to 50%) in 0.1% formic acid at 100 μ L/min with Thermo-Dionex Ultimate 3000 HPLC system, thus will lead to huge back-exchange. We updated above information in the revised manuscript.

How were the percentages in figure S6C computed? Was there a fully deuterated control? The mutagenesis helps alleviate some of this concern, but as the HDX was the basis of those mutations, and is still used to guide modeling, all of the associated methods and analysis of the HDX is something that really needs to be clarified and explained.

Response: The percentages in Fig. S7C were computed with a fully deuterated control. The fully deuterated samples were prepared by incubating protein with deuterium-exchange solution at room temperature for 24 hours, and then quench the reaction with 36 μ l of optimal quench solution. We supplemented above information in the revised manuscript.

5) Minor issue: I'm guessing the column used for HDX-MS was a 1x50mm (millimeter, not microMolar).

Response: We corrected this in the revised manuscript.

REVIEWERS' COMMENTS

Reviewer #1 (Remarks to the Author):

all my comments suggestions were addressed in full

Reviewer #2 (Remarks to the Author):

All my comments have been addressed. The manuscript is ready for publication.

Reviewer #3 (Remarks to the Author):

The authors addressed several comments, however they did not include the spreadsheet for reporting critical HDX-MS information, along with the summary spreadsheet of the kinetics data as outlined recently in: "Recommendations for performing, interpreting and reporting hydrogen deuterium exchange mass spectrometry (HDX-MS) experiments" *Nature Methods*, 16: 595–602 (2019), which provides template spreadsheets that the authors can use to assemble the necessary additional material. If the authors are using HDEaminer then that software has built-in features to generate the summary of the kinetics data and provides all the metrics necessary for the spreadsheet of the critical HDX-MS information (coverage, conditions, redundancy, back exchange levels, thresholds for significance, etc). The authors will need to provide these two items as supplementary data prior to publication.

Last minor note: HDEaminer is not from Thermo, it is from Sierra Analytics.

Reviewer #3 (Remarks to the Author):

The authors addressed several comments, however they did not include the spreadsheet for reporting critical HDX-MS information, along with the summary spreadsheet of the kinetics data as outlined recently in: “Recommendations for performing, interpreting and reporting hydrogen deuterium exchange mass spectrometry (HDX-MS) experiments” Nature Methods, 16: 595–602 (2019), which provides template spreadsheets that the authors can use to assemble the necessary additional material. If the authors are using HDExaminer then that software has built-in features to generate the summary of the kinetics data and provides all the metrics necessary for the spreadsheet of the critical HDX-MS information (coverage, conditions, redundancy, back exchange levels, thresholds for significance, etc). The authors will need to provide these two items as supplementary data prior to publication.

Response: As suggested, we have included the kinetics data summary in Supplementary Information Table S6 and provided all the metrics data on the critical HDX-MS information in an Excel file named “HDX_table”.

Last minor note: HDExaminer is not from Thermo, it is from Sierra Analytics.

Response: We have corrected this.